# Probably Approximately Precision and Recall Learning

**Lee Cohen**[*]
Stanford
leecohencs@gmail.com

**Yishay Mansour**
Tel Aviv University and Google Research
mansour.yishay@gmail.com

**Shay Moran**
Technion and Google Research
smoran@technion.ac.il

**Han Shao** [†]
University of Maryland, College Park
hanshao@umd.edu

## Abstract

*Precision* and *Recall* are fundamental metrics in machine learning tasks where both accurate predictions and comprehensive coverage are essential, such as in multi-label learning, language generation, medical studies, and recommender systems. A key challenge in these settings is the prevalence of one-sided feedback, where only positive examples are observed during training—e.g., in multi-label tasks like tagging people in Facebook photos, we may observe only a few tagged individuals, without knowing who else appears in the image. To address learning under such partial feedback, we introduce a Probably Approximately Correct (PAC) framework in which hypotheses are set functions that map each input to a set of items, extending beyond single-label predictions and generalizing classical binary, multi-class, and multi-label models. Our results reveal sharp statistical and algorithmic separations from standard settings: classical methods such as Empirical Risk Minimization provably fail, even for simple hypothesis classes. We develop new algorithms that learn from positive data alone, achieving optimal sample complexity in the realizable case, and establishing multiplicative—rather than additive—approximation guarantees in the agnostic case, where achieving additive regret is impossible.

## 1 Introduction

*Precision* and *Recall* are fundamental metrics in many machine learning applications, including multi-label learning, medical studies, generative models, information retrieval, and recommender systems, where the goal is to learn a set for each input and both accurate prediction and comprehensive coverage are critical. For example, in multi-label learning—where each input may correspond to multiple labels (e.g., an image containing multiple objects or a group photo containing multiple people)—the objective is to return the complete set of labels associated with each input. Recommender systems also rely heavily on precision and recall; for instance, Netflix aims to recommend a list of shows that includes all those a user would enjoy and excludes those they would not. In medical studies, precision, called *positive predictive value* (PPV), and recall, called *sensitivity*, are key performance metrics. For instance, if a test predicts that 10 patients have a condition and 9 truly have it, its PPV (precision) is $90\%$. If 5 patients actually have the condition and the test correctly identifies 4 of them, its sensitivity (recall) is $80\%$. In language identification and generation in the limit Gold (1967);

---

[*]Authors are ordered alphabetically.
[†]Work done while at Harvard.

Kleinberg & Mullainathan (2024); Kalavasis et al. (2024), *hallucination* denotes a failure of precision, where the model produces strings outside the true language, whereas *mode collapse* denotes a failure of recall, where the model's outputs become so constrained that valid, previously unseen strings are never generated .

A critical aspect of designing such systems is balancing two key metrics:

- Precision — the proportion of output items that are in the ground-truth set. Low *precision loss* means that most output items are correct.
- Recall — the proportion of ground-truth items that are included in the output. Low *recall loss* means the system captures most of the correct items.

Precision and recall are often at odds. Increasing the size of output set can improve recall but may reduce precision. In an ideal scenario, we might consider a full information model where, for each input in the training data, the algorithm has access to the set of *all* the ground-truth items. This setup implicitly includes negative examples, as items not in the set are known to be incorrect. At test time, the algorithm would then predict a set of items for each given input. Such a setting aligns well with the standard *Probably Approximately Correct* (PAC) framework Valiant (1984), allowing us to apply standard PAC solutions (e.g., Empirical Risk Minimization).

However, the assumption of full information is often unrealistic in many applications. In real-world scenarios, we typically observe only a small fraction of the ground-truth items (e.g., some objects in an image) during training, with no explicit information about unobserved items (e.g., which objects are not in the image). This situation is better characterized as a partial-information model. For instance, on Facebook, for each photo in the training data, we might only know some of the tagged individuals, without knowing whether any untagged person appears in the photo or not. To formalize this setting, we consider a simple abstraction: during training, for each input, we observe only *a single item sampled uniformly at random* from its ground-truth set. At test time, given a random input, the model is expected to return the full set of associated items, not just one.

Following standard practice in learning theory, given an input space $\mathcal{X}$ and a label space $\mathcal{Y}$, we consider a hypothesis class $\mathcal{H}$. Each hypothesis is a set function $g : \mathcal{X} \to 2^{\mathcal{Y}}$ that maps each input in $\mathcal{X}$ to a set of labels in $\mathcal{Y}$. For example, in the context of multi-label learning, inputs may be images and items may be objects; a hypothesis maps an image to the set of objects it contains. Our goal is to find a set function that minimizes both precision and recall losses. These losses are defined with respect to a target hypothesis $g^{\text{target}}$, which captures the ground-truth label sets, using the standard counting measure:

$$\ell^{\text{precision}}(g) := \mathbb{E}_{x \sim \mathcal{D}} \left[ \frac{|g(x) \setminus g^{\text{target}}(x)|}{|g(x)|} \right] \quad \text{and} \quad \ell^{\text{recall}}(g) := \mathbb{E}_{x \sim \mathcal{D}} \left[ \frac{|g^{\text{target}}(x) \setminus g(x)|}{|g^{\text{target}}(x)|} \right] ,$$

where $\mathcal{D}$ denotes the distribution over inputs. Precision and recall are equal to 1 minus the precision and recall losses, respectively. Binary and multiclass PAC learning can be viewed as special cases of our model, where we restricted the set size to be 1. Note that, for each given $x$, we adopt the standard counting measure to quantify the difference between $g(x)$ and $g^{\text{target}}(x)$. While other measures are possible, it is often challenging to define more sophisticated alternatives in a principled and intuitive way. To the best of our knowledge, the only generalization of precision and recall beyond counting measure is proposed by Sajjadi et al. (2018), though the resulting notions are relatively complex. Hence, we will be focusing on the counting measure in this paper and leave a more general notion of precision and recall as a open question.

We distinguish between two settings- *realizable* and *agnostic*. In the realizable setting, we assume the target hypothesis $g^{\text{target}}$ is in the class $\mathcal{H}$, and, aim to find a hypothesis with small precision and recall losses. In the agnostic setting, we do not assume that a perfect hypothesis is in the class. Instead, we aim to compete with the "best" hypothesis in the class, acknowledging that some error is unavoidable. In the context of the agnostic setting, defining the "best" hypothesis is subtle. One hypothesis might have high precision but low recall, while another has the opposite. Depending on the application's needs, one may prefer higher precision over recall or vice versa.

This naturally leads us to the concept of *Pareto-loss* objective, which captures the trade-offs between precision and recall along the *Pareto frontier*.[3]

---

[3]The term "Pareto" originates from Vilfredo Pareto, an economist who observed that certain distributions followed a pattern where improvements in one dimension often involved trade-offs in another. The concept

Namely, the "best" hypotheses are on a Pareto frontier, which is the set of hypotheses where no other hypotheses are better in both precision and recall simultaneously (see, e.g., Figure 1). Here, we try, given desired precision and recall losses parameters $(p, r)$ to return a hypothesis whose precision and recall losses $(p', r')$ satisfy $p' \lesssim p$ and $r' \lesssim r$. For example, one might aim to optimize precision while keeping recall below a specific threshold (e.g., at most $0.5$).

Our goal is to design algorithms whose sample complexity is polynomial in the log size of the class and the inverses of the accuracy and confidence parameters, and in some cases, on the maximum size of the output (which is arguably small in certain applications). We focus on finite hypothesis classes and aim for sample complexity bounds that depend logarithmically, rather than linearly, on the size of the class. This goal is motivated by standard sample complexity bounds in PAC learning and is particularly relevant when training data is costly to obtain, as is often the case with human-provided feedback.[4]

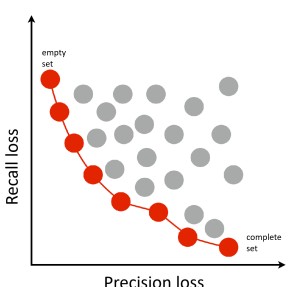

Figure 1: Example of hypotheses with varying precision and recall losses. Each point is a distinct hypothesis, with red points on the Pareto frontier, showing optimal trade-offs between precision and recall losses. The empty set function (which always return the empty set) always achieves zero precision loss (but has no guarantee on the recall loss), while the complete set function (which always return the entire label space $\mathcal{Y}$ for any given input $x$) always achieves zero recall loss (but has no guarantee on the precision loss).

Our learning problem is significantly more challenging than standard supervised learning tasks due to the absence of negative examples—that is, input-label pairs where the label is not in the ground-truth set. We observe only positive examples, making it impossible to estimate precision loss directly: without knowing which labels are incorrect, we cannot determine how many incorrect labels a hypothesis might output. This limitation undermines standard supervised learning approaches, which rely on access to both positive and negative examples.

In standard supervised learning, a classical solution known as *Empirical Risk Minimization* (ERM) involves finding a hypothesis that best fits the data by minimizing the average loss over all observed examples. However, in our case, the absence of negative examples means that ERM cannot be applied effectively, as there is no way to determine how well a hypothesis avoids incorrect labels. For example, the complete set function (where $g(x) = \mathcal{Y}$ outputs all items) is consistent with every training set, since we have no negative examples to contradict it. However, such a hypothesis might have poor precision. Without negative examples in the training set, any hypothesis that covers all observed positive examples appears equally valid, even if it recommends many irrelevant items and incurs a high precision loss. The failure of the ERM principle is not unique to our learning problem; it also occurs in other learning problems, such as multi-class classification Daniely et al. (2015), density estimation Devroye & Lugosi (2001); Bousquet et al. (2019), and partial concept learning Alon et al. (2022).

In fact, it is not just that ERM would fail; we provide an example in which two hypotheses have nearly the same recall loss but very different precision losses, making it impossible to determine which hypothesis has better precision loss based solely on the training data (see Example 1 for more details). These challenges necessitate novel approaches to learning and evaluating hypotheses.

**Our Contributions**

- **Learning Model:** We propose a learning model that operates under partial information, where the algorithm observes only positive examples. For each input drawn from unknown distribution $\mathcal{D}$, it observes one label sampled uniformly at random from its ground-truth set. This model reflects

---

of Pareto frontier, inspired by this principle, represents optimal trade-offs between competing objectives. Our Pareto loss captures the balance between precision and recall, aiming to improve both while acknowledging the trade-off.

[4]While it would be interesting to explore infinite hypothesis classes and characterize them via a combinatorial measure in the style of VC dimension, the findings in Lechner & Ben-David (2024) suggest that such a dimension may not exist in this setting. We leave this as an intriguing open question, as addressing it falls beyond the scope of this work, which focuses on the finite case—a setting that is already challenging.

real-world constraints and is more practical than assuming access to complete ground-truth set as done in multi-label learning.

- **Realizable Setting:** We design algorithms that, given a sample of size

$$O\left(\frac{\log(|\mathcal{H}|/\delta)}{\varepsilon}\right),$$

achieve recall and precision losses of at most $\varepsilon$ with probability at least $1 - \delta$. We propose two distinct approaches to achieve this goal: the first circumvents the challenge of estimating precision by minimizing an appropriate surrogate loss, while the second takes a more intuitive approach inspired by the maximum likelihood principle Shalev-Shwartz & Ben-David (2014). In essence, this second algorithm, when presented with two hypotheses consistent with the data, prioritizes the one with smaller output sizes in a suitably quantified sense.
- **Agnostic Setting:** We demonstrate that achieving a vanishing additive error, as is standard in learning theory, is impossible in this setting by providing lower bounds on the sum of precision and recall losses, with multiplicative factors greater than 1. In the other direction, we show that constant multiplicative factor guarantees are indeed achievable by adapting our realizable setting algorithms to the agnostic case. Closing the gap between our upper and lower bounds on the best achievable multiplicative factor (5 vs. 1.05) remains an open question. For the Pareto-loss case, we establish both upper and lower bounds for the following question: Given that there exists a hypothesis in the class with precision and recall losses $(p, r)$, which guarantee pairs $(p', r')$ are achievable? Finally, we pose open questions about the optimal factors in the agnostic setting.

**Related Work** Precision and recall are natural and standard metrics used broadly in machine learning, spanning applications from binary classification (Juba & Le, 2019; Diochnos & Trafalis, 2021), multi-class classification (Grandini et al., 2020), regression (Torgo & Ribeiro, 2009), and time series (Tatbul et al., 2018) to information retrieval (Arora et al., 2016) and generative models (Sajjadi et al., 2018). Beyond precision-recall, another related metric—the area under the ROC curve (AUC)—has also been extensively studied in the history of binary classification (Cortes & Mohri, 2003, 2004; Rosset, 2004; Agarwal et al., 2005), with a focus on generalization. Our work, however, studies a different problem of multi-label learning where the goal is to output a list of labels for each input. In the context of recommender systems, recommending a list of items has also been addressed in cascading bandits (Kveton et al., 2015). However, while our objective is to identify the items that each input likes, their focus is on learning the top $K$ items that are liked by most inputs. Another feature of our learning problem is that we can only learn from positive examples. PAC learning for binary classification from positive examples has been studied in the literature (Denis, 1998; De Comité et al., 1999; Letouzey et al., 2000; Bekker & Davis, 2020).

Multi-label learning (McCallum, 1999; Schapire & Singer, 2000) has been an area of study in machine learning, with various, primarily experimental approaches (see, e.g., (Elisseeff & Weston, 2001; Petterson & Caetano, 2011; Kapoor et al., 2012) and (Zhang & Zhou, 2014; Bogatinovski et al., 2022) for surveys). In multi-label learning, the training set consists of examples, each associated with multiple labels rather than just one. The goal is to train a model that can learn the relationships between the features of each example and all its labels. At test time, the learner predicts a list of labels for new examples, aiming to capture all the relevant labels, rather than just a single one.

Some works have examined multi-label learning from a theoretical standpoint, focusing in particular on the Bayes consistency of surrogate losses. Bayes-consistency in multi-label learning ensures that minimizing a surrogate loss also leads to minimizing the true target loss, which is crucial in multi-label settings where optimizing the actual loss is often computationally infeasible as it is non-convex, discrete losses in multi-label settings. Initiated by Gao & Zhou (2011) who first addressed Bayes-consistency for Hamming and ranking losses, showing binary relevance's consistency with Hamming loss but highlighting ranking loss difficulties. Extensions include rank-based metrics like precision@$\kappa$ and recall@$\kappa$ (Menon et al., 2019), which are loss functions defined under the constraint that the number of labels predicted by the model is limited to $\kappa$. Recently, Mao et al. (2024) established $H$-consistency bounds for multi-label learning, offering stronger guarantees than Bayes-consistency by providing non-asymptotic guarantees that apply to finite number of samples. Our model is inherently more challenging than traditional multi-label learning because our training set consists of examples, each associated with only a single correct label rather than all possible correct labels, with no negative feedback. Yet, at test time, the learner still needs to predict a list of relevant labels for new examples.

## 2 Model

As is standard in learning theory, we assume a hypothesis class $\mathcal{H}$ of set functions, our goal is to design algorithms whose sample complexity is polynomial in the log size of the class and the inverses of the accuracy and confidence parameters. More specifically, we are given a input space $\mathcal{X}$, a label space $\mathcal{Y}$ and a hypothesis class $\mathcal{H}$, in which every hypothesis $h : \mathcal{X} \mapsto 2^{\mathcal{Y}}$ maps each $x$ to a set of labels in $\mathcal{Y}$. We denote an unknown target hypothesis $g^{\text{target}}$. The training set consists of a sequence $(x_i, v_i)_{i=1}^m$ with $(x_i, v_i) \in \mathcal{X} \times \mathcal{Y}$. The inputs $x_1, \ldots, x_m$ are drawn IID from unknown distribution $\mathcal{D}$. For each input $x_i$, a random label $v_i$ is drawn uniformly from its ground-truth set, $g^{\text{target}}(x_i)$ defined by the target hypothesis. The algorithm then outputs a hypothesis $g^{\text{output}}$, and the goal is to minimize the expected precision and recall losses:

$$\ell^{\text{precision}}(g^{\text{output}}) := \mathbb{E}_{x \sim \mathcal{D}} \left[ \frac{|g^{\text{output}}(x) \setminus g^{\text{target}}(x)|}{n_{g^{\text{output}}}(x)} \right] ,$$

$$\ell^{\text{recall}}(g^{\text{output}}) := \mathbb{E}_{x \sim \mathcal{D}} \left[ \frac{|g^{\text{target}}(x) \setminus g^{\text{output}}(x)|}{n_{g^{\text{target}}}(x)} \right] ,$$

where for any hypothesis $g$, $n_g(x) = |g(x)|$ denotes the set size of $g$ at $x$.

We focus on designing learning rules that can compete with the "best" hypothesis in $\mathcal{H}$. Specifically, if there is a hypothesis $g \in \mathcal{H}$ with precision and recall losses $p$ and $r$, respectively, can we output a hypothesis $g^{\text{output}}$ whose precision and recall losses are comparable to $p$ and $r$? To answer this question we consider two natural metrics: scalar loss and Pareto loss.

**Scalar-Loss Objective**  The scalar loss is defined as the average of precision and recall losses[5]

$$\ell^{\text{scalar}}(g) := \frac{\ell^{\text{precision}}(g) + \ell^{\text{recall}}(g)}{2} .$$

For any $\alpha > 0$, we say $\alpha$-approximate optimal scalar loss is achievable if there exists a polynomial $P$ such that, for any finite hypothesis class $\mathcal{H}$, there is an algorithm $\mathcal{A}$ such that the following holds: For any $\varepsilon, \delta > 0$ and any distribution $\mathcal{D}$, if $\mathcal{A}$ is given an IID training set of size $P(\log|\mathcal{H}|, 1/\varepsilon, 1/\delta)$, with probability at least $1 - \delta$, it outputs a hypothesis with scalar loss satisfying

$$\ell^{\text{scalar}}(g^{\text{output}}) \leq \alpha \cdot \min_{g \in \mathcal{H}} \ell^{\text{scalar}}(g) + \varepsilon .$$

Our primary focus is on finite hypothesis classes, where we discuss dependencies on the cardinality of the hypothesis class, as is common in standard learning theory. We emphasize that the average loss is a natural way to combine precision and recall losses. Other scalarizations, such as the F1 score, are also possible. Several results in this paper extend to alternative definitions of scalar loss functions, which we discuss in the Discussion section.

**Pareto-Loss Objective**  Let $p, p', r, r' \in [0, 1]$. We write $(p, r) \implies (p', r')$ to denote the following statement: there exists a polynomial $P$ such that, for any finite hypothesis class $\mathcal{H}$, there is an algorithm $\mathcal{A}$ such that the following holds: If $\mathcal{D}$ is a distribution for which there exists a hypothesis in $\mathcal{H}$ with precision and recall losses $(p, r)$, then for any $\varepsilon, \delta > 0$, if $\mathcal{A}$ is given $p, r$ and an IID training set of size $P(\log|\mathcal{H}|, 1/\varepsilon, 1/\delta)$, with probability at least $1 - \delta$, it outputs a hypothesis with precision and recall losses at most $p' + \varepsilon$ and $r' + \varepsilon$.[6]

We are asking the following a question for each of our losses:

> **What is the smallest $\alpha$ such that $\alpha$-approximate optimal scalar loss is achievable?**
> **Given $p, r \in [0, 1]$, which pairs $(p', r')$ satisfy $(p, r) \implies (p', r')$?**

This hypothesis learning problem is considerably more challenging than standard supervised learning. If the entire ground-truth set $g^{\text{target}}(x_i)$ were observed in the training set rather than a random label $v_i \sim \text{Unif}(g^{\text{target}}(x_i))$, the task would reduce to standard supervised learning. However, observing only a random label prevents an unbiased estimate of precision loss, complicating the problem. We demonstrate it in the following example.

---

[5]The results can be generalized to any weighted sum of precision and recall losses via $w_1 \ell^{\text{precision}}(g) + w_2 \ell^{\text{recall}}(g) \leq 2 \max(w_1, w_2) \ell^{\text{scalar}}(g)$.

[6]Actually, our algorithms only requires the knowledge of $r$.

**Example 1.** *In Fig 2, the target hypothesis associates input $x_i$ with a large number $n$ of labels. Our hypothesis class contains two hypotheses, each predicting exactly one label for input $x_i$. In the red hypothesis $g_1$, the predicted label is always a true positive (i.e., it belongs to the ground-truth set), while in the blue hypothesis $g_2$, the predicted label is a false positive (i.e., it is not in the ground-truth set). Both hypotheses incur high recall loss: $\ell^{recall}(g_1) = \frac{n-1}{n}$ and $\ell^{recall}(g_2) = 1$. However, they differ significantly in precision loss: $g_1$ has a precision loss of 0, as it always predicts a correct label, whereas $g_2$ has a precision loss of 1, as it always predicts an incorrect label. Since $n$ is large, the probability of observing either predicted label in the training data is very low, making it impossible to distinguish between these two hypotheses—despite their drastically different precision losses.*

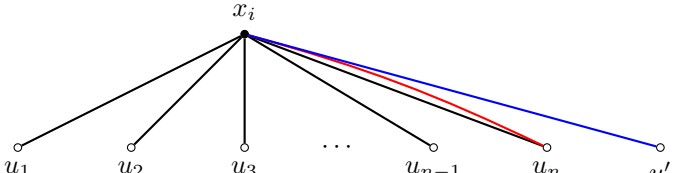

Figure 2: The target hypothesis (black) outputs $g^{\text{target}}(x_i) = \{u_1, \dots, u_n\}$ where $n$ is huge. The hypothesis $g_1$ (red) outputs only one label $u_n \in g^{\text{target}}(x_i)$ while $g_2$ (blue) outputs only one label $u' \notin g^{\text{target}}(x_i)$.

One might argue that the issue in the above example arises from the large size of the ground-truth set; however, we will later show that even when the ground-truth set has a small size, accurately estimating and optimizing precision remains impossible.

We emphasize that, unlike in other supervised learning settings, such as PAC learning, where minimizing empirical risk is often straightforward (e.g., by outputting any classifier that is consistent with the training set), here the learner only observes a single label $v_i$ per input $x_i$, rather than its entire ground-truth set. As a result, minimizing empirical precision loss in this context is far from trivial. For instance, regardless of the ground-truth set, a complete set function, which outputs the entire label space $\mathcal{Y}$ for any input, is always consistent with the training data but can still incur high precision loss.

## 3 Main Results

**The Realizable Setting**   We begin by presenting our results in the realizable setting, where the target hypothesis belongs to the hypothesis class $\mathcal{H}$. In this case, there is no distinction between optimizing the scalar-loss objective and the Pareto-loss objective. We propose two new algorithms that achieve both the scalar-loss and Pareto-loss objectives: one based on maximum likelihood estimation, and the other on minimizing a surrogate loss.

**Theorem 1.** *In the realizable setting, there exist algorithms such that given an IID training set of size $m \geq O(\frac{\log(|\mathcal{H}|/\delta)}{\varepsilon})$, with probability at least $1 - \delta$, the output hypothesis $g^{output}$ satisfies*

$$\ell^{recall}(g^{output}) \leq \varepsilon, \quad \ell^{precision}(g^{output}) \leq \varepsilon.$$

Since $\mathbb{1}(v_i \notin g(x_i))$ is an unbiased estimate of the recall loss $\ell^{\text{recall}}(g)$, any consistent hypothesis (i.e., hypothesis $g$ with $\sum_{i=1}^{m} \mathbb{1}(v_i \notin g(x_i)) = 0$) will have low recall loss. But ERM does not work as the training set contains only positive examples, and a complete set function is consistent with any training set but can incur high precision loss. Hence, the main challenge lies in minimizing the precision loss. Below are high-level descriptions of two algorithms designed to tackle this problem.

**Algorithm 1: Maximum Likelihood.**   One of our proposed algorithms is based on the natural idea of *maximum likelihood*. At a high level, although multiple hypotheses may be consistent with the training set, for any observed input-label pair $(x_i, v_i)$, if $v_i$ is contained in $g(x_i)$, the probability of observing this label is $\frac{1}{n_g(x_i)}$ when $g$ is the target hypothesis. The maximum likelihood method returns $g^{\text{output}} = \arg\max_{g \in \mathcal{H}} \prod_{i=1}^{m} \frac{\mathbb{1}(v_i \in g(x_i))}{n_g(x_i)}$, which is equivalent to returning the hypothesis with

the minimum sum of log output sizes among all consistent hypotheses, i.e.,

$$g^{\text{output}} = \arg\min_{g:g \text{ is consistent}} \sum_{i=1}^{m} \log(n_g(x_i)) \,.$$

Consequently, we can rule out the complete set function, as its sum of log output size $\sum_{i=1}^{m} \log(n_g(x_i))$ is huge. Any consistent hypothesis will have low empirical recall loss by applying standard concentration inequality and thus, $\widehat{\ell}^{\text{recall}}(g^{\text{output}})$ is small and we only need to show that the empirical precision loss is small. The main technical challenge in the analysis is *how to connect precision loss with likelihood.*

**Algorithm 2: Minimizing a Surrogate Loss.** The other algorithm is more directly aligned with the scalar-loss objective. While we cannot obtain an unbiased estimate of the precision loss, and hence the scalar loss, we introduce a surrogate loss that both upper- and lower-bounds the scalar loss within a constant multiplicative factor. Then we output a hypothesis minimizing this surrogate loss.

For any hypothesis $g$, we define a vector $v_g : \mathcal{H} \times \mathcal{H} \to [0, 1]$ by

$$v_g(g', g'') = \frac{1}{m} \sum_{i=1}^{m} \mathbb{P}_{v \sim \text{Unif}(g(x_i))} \left( v \in g'(x_i) \setminus g''(x_i) \right) .$$

Intuitively, $v_g(g', g'')$ represents the fraction of correct labels that are output by $g'$ but not by $g''$ in the counterfactual scenario where $g$ is the target hypothesis. If $g$ is indeed the target hypothesis, then this quantity should be consistent with our training data, $v_g(g', g'') \approx v_{\widehat{g}}(g', g'')$, where $\widehat{g}$ is the observed (empirical) hypothesis; i.e., the hypothesis in which every $x_i$ is only associated with the random label $v_i$ which is observed in the training set.

We then define a metric $d_{\mathcal{H}}$ between two hypotheses $g_1$ and $g_2$ by

$$d_{\mathcal{H}}(g_1, g_2) = \|v_{g_1} - v_{g_2}\|_\infty \,.$$

Surprisingly, we show that $d_{\mathcal{H}}(g^{\text{target}}, g)$ is a surrogate for the scalar loss, providing both lower and upper bounds on the scalar loss with a constant multiplicative factor.

**The Agnostic Setting** In the agnostic setting, we show in the next two theorems that it is impossible to achieve an additive error for both scalar-loss and Pareto-loss objectives as is standard in learning theory.

**Theorem 2.** *There exists a class $\mathcal{H} = \{g_1, g_2\}$ of two hypotheses, such that for any (possibly randomized improper) algorithm, there exists a target hypothesis $g^{target}$ with bounded output set size (that is, there exists a constant $C$ such that $|g^{target}(x)| \leq C$ for all $x \in \mathcal{X}$) and a data distribution $\mathcal{D}$ with $\min_{g \in \mathcal{H}}(\ell^{scalar}(g)) > 0$ s.t. for any sample size $m > 0$, with probability 1 over the training set, the expected (over the randomness of the algorithm) loss of the output $g^{output}$*

$$\mathbb{E}\left[\ell^{scalar}(g^{output})\right] \geq 1.05 \cdot \min_{g \in \mathcal{H}}(\ell^{scalar}(g)) \,.$$

**Theorem 3.** *There exists a class $\mathcal{H} = \{g_1, g_2\}$ of two hypotheses, such that for any (possibly randomized improper) algorithm given the knowledge of $(p, r) = (\frac{7}{16}, \frac{1}{4})$, there exists a target hypothesis $g^{target}$ with bounded output set size and a data distribution $\mathcal{D}$ for which there exists a hypothesis $g^\dagger \in \mathcal{H}$ with $\ell^{recall}(g^\dagger) = \frac{1}{4}$ and $\ell^{precision}(g^\dagger) = \frac{7}{16}$ s.t. for any sample size $m > 0$, with probability 1 over the training set, the expected (over the randomness of the algorithm) precision and recall losses of the output $g^{output}$ satisfy*

$$\mathbb{E}\left[\ell^{recall}(g^{output})\right] + \frac{12}{5} \mathbb{E}\left[\ell^{precision}(g^{output})\right] \geq \frac{7}{5} \,.$$

**Remark 1.** *Hence the output hypothesis either suffers $\ell^{recall}(g^{output}) > \frac{1}{4} = \ell^{recall}(g^\dagger)$ or $\ell^{precision}(g^{output}) \geq \frac{23}{48} = \frac{23}{21} \ell^{precision}(g^\dagger)$. Thus $(\frac{7}{16}, \frac{1}{4}) \not\Rightarrow (\frac{7}{16} + 0.01, \frac{1}{4} + 0.01)$.*

For any hypothesis $g$, it's precision loss at any input $x$ is $\frac{|g(x) \setminus g^{\text{target}}(x)|}{n_g(x)}$ and we only get a random label $v \sim \text{Unif}(g^{\text{target}}(x))$. If we are given the knowledge of the output size $n_{g^{\text{target}}}(x)$ of the target

hypothesis, then we can obtain an unbiased estimate of the precision loss, i.e., $1 - \frac{n_{g^{\text{target}}}(x)}{n_g(x)} \cdot \mathbb{1}(v \in g(x))$. But the difficulty lies in that we don't know $n_{g^{\text{target}}}(x)$.

Based on these two lower bounds, in the scalar-loss case, we allow for a multiplicative factor $\alpha$. In the Pareto-loss case, we ask a more general question: which pairs of guarantees $(p', r')$ are achievable, given that there exists a hypothesis in the class with precision and recall losses $(p, r)$? Since the recall loss is optimizable, for any given $r$, if there exists a hypothesis in the class with recall loss $r$, we can always achieve that recall loss. Therefore, we refine our question as follows: given any $p, r \in [0, 1]$, what is the minimum precision loss $p'$ such that $(p, r) \Rightarrow (p', r)$?

Since the recall is estimable, when we have an algorithm achieving $\alpha$-approximate scalar loss, we can first eliminate all hypotheses with recall loss higher than $r$ and then run this algorithm over the remaining hypotheses. Then we can achieve $(p, r) \Rightarrow (\alpha(p + r), r)$.

**Theorem 4.** *There exist an algorithm such that given an IID training set of size $m \geq O(\frac{\log(|\mathcal{H}|/\delta)}{\varepsilon^2})$, with probability at least $1 - \delta$, the output hypothesis $g^{output}$ satisfies*

$$\ell^{scalar}(g^{output}) \leq 5 \cdot \min_{g \in \mathcal{H}} \ell^{scalar}(g) + \varepsilon \,.$$

*This implies that for any $p, r \in [0, 1]$, $(p, r) \Rightarrow (5(p + r), r)$.*

This result is achieved using the same surrogate loss idea in the realizable setting. However, in the agnostic setting, applying maximum likelihood directly no longer works. This is because it is possible that none of the hypotheses in the hypothesis class are consistent with the training set and thus all hypotheses have zero likelihood. Instead, we make some modifications to adapt the maximum likelihood idea work for Pareto-loss objective.

As mentioned earlier, the maximum likelihood method is equivalent to returning the hypothesis with the minimum sum of log output sizes among all consistent hypotheses. Hence, it can be decomposed into two steps: minimizing the recall loss and then regulating by minimizing the sum of log output sizes. In the agnostic setting, given any $r$, we first find the set $\widehat{\mathcal{H}}$ of all hypotheses in the hypothesis class with recall loss at most $r + 2\varepsilon$, and then regulate by minimizing the sum of log truncated output sizes by returning

$$g^{output} = \arg\min_{g' \in \widehat{\mathcal{H}}} \max_{g \in \widehat{\mathcal{H}}} \frac{1}{m} \sum_{i \in [m]} \log \frac{n_{g'}(x_i) \wedge 4n_g(x_i)}{n_g(x_i) \wedge 4n_{g'}(x_i)} \,,$$

where $a \wedge b := \min(a, b)$. The truncation plays an important role here. Intuitively, let $g^\dagger$ denote the hypothesis with precision and recall losses $p$ and $r$, respectively. If there exists an $x_i$ such that $n_{g^\dagger}(x_i)$ is very large, minimizing the sum of log-untruncated output sizes will never return $g^\dagger$. By applying truncation, we limit the effect of a single input with a very large output size.

As we can see, there is a gap between the upper and lower bounds in the agnostic case, leaving an open question: What is the optimal multiplicative approximation factor $\alpha$ in the scalar case, and what is the optimal $p'$ such that $(p, r) \Rightarrow (p', r')$?

**The Semi-Realizable Setting**    The results in the agnostic setting fail to offer meaningful guarantees in certain natural scenarios, such as $p = 0$ and $r = \frac{1}{2}$ (i.e., when there exists a hypothesis that captures half of ground-truth labels without outputting any incorrect labels). We are therefore interested in the following question: whether $(p = 0, r) \Rightarrow (p' = 0, r' = r)$?

We propose an algorithm with sample complexity depending on output set size of the target hypothesis and show that it is impossible to achieve zero precision loss with sample complexity independent of the target hypothesis's output set size.

As discussed previously, it's impossible for us to estimate the precision loss. However, we can still separate hypotheses with zero precision loss and non-zero precision loss if the the target hypothesis's output set size is bounded. If the precision loss of hypothesis $g$ at input $x$ is 0, i.e.,

$$\ell^{precision}(g, x) = \frac{|g(x) \setminus g^{\text{target}}(x)|}{n_g(x)} = 1 - \frac{|g(x) \cap g^{\text{target}}(x)|}{n_g(x)} = 0 \,,$$

we have

$$\frac{|g(x) \cap g^{\text{target}}(x)|}{n_g(x) \cdot n_{g^{\text{target}}}(x)} = \frac{1 - \ell^{precision}(g, x)}{n_{g^{\text{target}}}(x)} = \frac{1}{n_{g^{\text{target}}}(x)} \,.$$

If the precision loss $\ell^{\text{precision}}(g, x)$ is positive, we have

$$\frac{|g(x) \cap g^{\text{target}}(x)|}{n_g(x) \cdot n_{g^{\text{target}}}(x)} = \frac{1 - \ell^{\text{precision}}(g, x)}{n_{g^{\text{target}}}(x)} < \frac{1}{n_{g^{\text{target}}}(x)} .$$

Hence, we can use the quantity $\mathbb{E}_{x \sim \mathcal{D}}\left[\frac{|g(x) \cap g^{\text{target}}(x)|}{n_g(x) \cdot n_{g^{\text{target}}}(x)}\right]$ to separate hypotheses with zero and non-zero precision loss, and it is estimable. For each hypothesis $g$, when the gap of this quantity between hypothesis with zero precision loss and $g$ is $\Delta_{g, \mathcal{D}} := \mathbb{E}_{x \sim \mathcal{D}}\left[\frac{1}{n_{g^{\text{target}}}(x)}\right] - \mathbb{E}_{x \sim \mathcal{D}}\left[\frac{|g(x) \cap g^{\text{target}}(x)|}{n_g(x) \cdot n_{g^{\text{target}}}(x)}\right] > 0$, then by obtaining $\frac{1}{\Delta_{g, \mathcal{D}}^2}$ samples of $x$, we can tell that $g$ has nonzero precision loss. Let $\Delta_{\mathcal{D}}$ be the smallest gap of this quantity between hypotheses with zero and nonzero precision losses:

$$\Delta_{\mathcal{D}} = \min_{g \in \mathcal{H}: \ell^{\text{precision}}(g) > 0} \Delta_{g, \mathcal{D}} .$$

Then we can have sample complexity dependent on this gap.

**Theorem 5.** *There exists an algorithm such that if there exists a hypothesis $g' \in \mathcal{H}$ with $\ell^{\text{precision}}(g') = 0$ and $\ell^{\text{recall}}(g') = r$, then given an IID training set of size $O(\frac{\log(|\mathcal{H}|/\delta)}{\Delta_{\mathcal{D}}^2})$, with probability $1 - \delta$, it outputs a hypothesis with $\ell^{\text{precision}}(g^{\text{output}}) = 0$ and $\ell^{\text{recall}}(g^{\text{output}}) = r$.*

When the target hypothesis's output size $n_{g^{\text{target}}}(x)$ is bounded by $C$ almost everywhere, we have

$$\mathbb{E}\left[\frac{|g(x) \cap g^{\text{target}}(x)|}{n_g(x) \cdot n_{g^{\text{target}}}(x)}\right] = \mathbb{E}\left[\frac{1 - \ell^{\text{precision}}(g, x)}{n_{g^{\text{target}}}(x)}\right] \leq \mathbb{E}\left[\frac{1}{n_{g^{\text{target}}}(x)}\right] - \mathbb{E}\left[\frac{\ell^{\text{precision}}(g, x)}{C}\right] .$$

Therefore, we have $\Delta_{g, \mathcal{D}} \geq \frac{\ell^{\text{precision}}(g)}{C}$. This implies that when the target hypothesis has bounded output size, we are able to find the hypothesis with zero precision loss. However, when the target hypothesis's output size becomes too large, we show that it is impossible to achieve precision-recall of $(0, r)$.

**Theorem 6.** *There exists a class $\mathcal{H} = \{g_1, g_2\}$ of two hypotheses, for any $m > 0$ and any (possibly randomized improper) algorithm $\mathcal{A}$, there exists a target hypothesis $g^{\text{target}}$ and a data distribution $\mathcal{D}$ for which there exists a hypothesis $g^\dagger \in \mathcal{H}$ with $\ell^{\text{precision}}(g^\dagger) = 0$ s.t. with probability $1 - \delta$ over the training set, the expected (over the randomness of the algorithm) precision and recall losses of the output $g^{\text{output}}$ satisfy either $\mathbb{E}\left[\ell^{\text{recall}}(g^{\text{output}})\right] \geq \min_{g \in \mathcal{H}} \ell^{\text{recall}}(g) + \Omega(1)$ or $\mathbb{E}\left[\ell^{\text{precision}}(g^{\text{output}})\right] = \Omega(1)$.*

In the proof of the theorem, we construct a target hypothesis with output size much larger than the sample size $m$, as well as hypotheses $g_1$ and $g_2$ with output size 1. In this setup, regardless of whether $g_1$ has perfect precision (i.e., the output of $g_1$ is a subset of the ground-truth set) or very poor precision, we cannot distinguish between these two cases because we never observe a label in the output of $g_1$ being sampled. Therefore, the only way to achieve zero precision loss is to output the empty set function, which, however, results in a high recall loss.

## 4 Discussion

One of the main goals of this paper is to study the Pareto front of precision and recall, which is reflected in our Pareto-loss objective. There are other scalar measures of predictive performance, the $F_1$ and the $F_\beta$ scores. $F_1$ and $F_\beta$ are direct functions of precision and recall, which is weaker than our Pareto-loss objective, and thus, the hardness results can be extended directly. We also note that in the realizable setting, we can minimize other alternatives for scalar losses as we can optimize both precision and recall simultaneously. Nevertheless, hypotheses with optimal $F_1$ or $F_\beta$ scores are specific points on this Pareto front, and it will be interesting for future work to find them efficiently.

Beyond that, there are two natural open questions. First, there is a gap between the upper and lower bounds. For the scalar-loss objective, we demonstrate that an $\alpha = 5$ approximate optimal scalar loss is achievable, while $\alpha = 1.05$ is not, leaving it unclear what the optimal $\alpha$ is. For the Pareto-loss objective, we establish an upper bound of $(p, r) \Rightarrow (5(p + r), r)$ and a lower bound of $(p, r) \not\Rightarrow (p + 0.01, r + 0.01)$, again suggesting a gap that we do not yet know how to close.

Second, it remains an open question whether there exists a combinatorial measure, similar to the VC dimension in standard PAC learning, that characterizes the learnability of precision and recall. Each hypothesis implicitly defines a distribution at each node—specifically, a uniform distribution over its neighborhood. In Section B.3, we also link the scalar loss to the total variation distance, thus reducing the scalar loss learning problem to a special case of distribution learning. However, as shown in Lechner & Ben-David (2024), there is no such a dimension characterizing the sample complexity of learning certain distribution classes (in their case, a mixture of point mass and uniform distributions). This result suggests a potential limitation in identifying a combinatorial measure for our learning problem.

## Acknowledgements

Lee Cohen is supported by the Simons Foundation Collaboration on the Theory of Algorithmic Fairness, the Sloan Foundation Grant 2020-13941, and the Simons Foundation investigators award 689988.

Yishay Mansour was supported by funding from the European Research Council (ERC) under the European Union's Horizon 2020 research and innovation program (grant agreement No. 882396), by the Israel Science Foundation, the Yandex Initiative for Machine Learning at Tel Aviv University and a grant from the Tel Aviv University Center for AI and Data Science (TAD).

Shay Moran is a Robert J. Shillman Fellow; he acknowledges support by ISF grant 1225/20, by BSF grant 2018385, by Israel PBC-VATAT, by the Technion Center for Machine Learning and Intelligent Systems (MLIS), and by the the European Union (ERC, GENERALIZATION, 101039692). Views and opinions expressed are however those of the author(s) only and do not necessarily reflect those of the European Union or the European Research Council Executive Agency. Neither the European Union nor the granting authority can be held responsible for them.

Han Shao was supported by Harvard CMSA.

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

# A Proof Overview

Given a sequence of IID inputs $x_1, \ldots, x_m$, let

$$\widehat{\ell}^{\text{precision}}(g) = \frac{1}{m} \sum_{i=1}^{m} \frac{|g(x_i) \setminus g^{\text{target}}(x_i)|}{n_g(x_i)}$$

and

$$\widehat{\ell}^{\text{recall}}(g) = \frac{1}{m} \sum_{i=1}^{m} \frac{|g^{\text{target}}(x_i) \setminus g(x_i)|}{n_{g^{\text{target}}}(x_i)}$$

denote the empirical precision and recall losses. It suffices to focus on empirical precision and recall losses minimization since by standard concentration bounds, minimizing these empirical losses leads to the minimization of the expected recall and precision losses.

**Minimizing Precision Loss Through Maximum Likelihood** The maximum likelihood method returns $g^{\text{output}} = \arg\max_{g \in \mathcal{H}} \prod_{i=1}^{m} \frac{\mathbb{1}(v_i \in g(x_i))}{n_g(x_i)}$, which is equivalent to returning the hypothesis with the minimum sum of log output sizes among all consistent hypotheses, i.e.,

$$g^{\text{output}} = \arg\min_{g : g \text{ is consistent}} \sum_{i=1}^{m} \log(n_g(x_i)).$$

Any consistent hypothesis will have low empirical recall loss by applying standard concentration inequality and thus, $\widehat{\ell}^{\text{recall}}(g^{\text{output}})$ is small and we only need to show that the empirical precision loss is small. The main technical challenge in the analysis is *how to connect precision loss with likelihood*.

Since $g^{\text{target}}$ is contained in the hypothesis class in the realizable setting and it is consistent with the training data, due to our algorithm, we have

$$\sum_{i=1}^{m} \log(n_{g^{\text{output}}}(x_i)) \leq \sum_{i=1}^{m} \log(n_{g^{\text{target}}}(x_i)). \tag{1}$$

We first prove that for any hypothesis $g$, its empirical precision loss can be bounded by a term of log output size and the empirical recall loss as follows:

$$\widehat{\ell}^{\text{precision}}(g) \leq \frac{2}{m} \sum_{i \in [m] : \frac{n_g(x_i)}{n_{g^{\text{target}}}(x_i)} \geq 1} \log \frac{n_g(x_i)}{n_{g^{\text{target}}}(x_i)} + 2\widehat{\ell}^{\text{recall}}(g). \tag{2}$$

By combining Eq (1) and (2), we have

$$\widehat{\ell}^{\text{precision}}(g^{\text{output}}) \leq -\frac{2}{m} \sum_{i : \frac{n_{g^{\text{output}}}(x_i)}{n_{g^{\text{target}}}(x_i)} < 1} \log \frac{n_{g^{\text{output}}}(x_i)}{n_{g^{\text{target}}}(x_i)} + 2\widehat{\ell}^{\text{recall}}(g^{\text{output}}).$$

However, with high probability, the first term $-\frac{2}{m} \sum_{i : \frac{n_{g^{\text{output}}}(x_i)}{n_{g^{\text{target}}}(x_i)} < 1} \log \frac{n_{g^{\text{output}}}(x_i)}{n_{g^{\text{target}}}(x_i)}$ is small. This is because, for any hypothesis $g$, the probability of outputting $g$ is

$$\mathbb{P}_{v_{1:m}}\left(g^{\text{output}} = g\right) \leq \mathbb{P}_{v_{1:m}}\left(g \text{ is consistent}\right) \leq \prod_{i : \frac{n_g(x_i)}{n_{g^{\text{target}}}(x_i)} < 1} \frac{n_g(x_i)}{n_{g^{\text{target}}}(x_i)}.$$

For any hypothesis $g$ with large $-\frac{2}{m} \sum_{i : \frac{n_g(x_i)}{n_{g^{\text{target}}}(x_i)} < 1} \log \frac{n_g(x_i)}{n_{g^{\text{target}}}(x_i)}$, the probability of outputting such a hypothesis is low.

**Modifying Maximum Likelihood in the Agnostic Setting**   In the agnostic setting, all hypotheses in the hypothesis class may have zero likelihood of being the true hypothesis; thus, the standard maximum likelihood method doesn't work. However, we can make slight modifications to the maximum likelihood method to make it work for the Pareto-loss objective.

As mentioned earlier, the maximum likelihood method is equivalent to returning the hypothesis with the minimum sum of log output sizes among all consistent hypotheses. Hence, it can be decomposed into two steps: minimizing the recall loss and then regulating by minimizing the sum of log output sizes. In the agnostic setting, given any $r$, we first find the set $\widehat{\mathcal{H}}$ of all hypotheses in the hypothesis class with recall loss at most $r + 2\varepsilon$, and then regulate by minimizing the sum of log truncated output sizes by returning

$$g^{\text{output}} = \arg\min_{g' \in \widehat{\mathcal{H}}} \max_{g \in \widehat{\mathcal{H}}} \frac{1}{m} \sum_{i \in [m]} \log \frac{n_{g'}(x_i) \wedge 4n_g(x_i)}{n_g(x_i) \wedge 4n_{g'}(x_i)},$$

where $a \wedge b := \min(a, b)$. The truncation plays an important role here. Intuitively, let $g^\dagger$ denote the hypothesis with precision and recall losses $p$ and $r$, respectively. If there exists an $x_i$ such that $n_{g^\dagger}(x_i)$ is very large, minimizing the sum of log-untruncated output sizes will never return $g^\dagger$. By applying truncation, we limit the effect of a single input with a very large output size.

**Minimizing the Surrogate for the Scalar Loss**   Here we consider an alternative learning rule based on two simple principles for discarding sub-optimal hypotheses. We illustrate these principles with the following intuitive example: consider a music recommender system, and assume we are considering two candidate hypotheses, $g'$ and $g''$. Both hypotheses recommend classical music; however, $g'$ recommends pieces by Bach 20% of the time and pieces by Mozart 10% of the time, while $g''$ never recommends any pieces by Mozart or Bach.

Now, suppose that in the training set, users frequently choose to listen to pieces by Mozart. This observation suggests that $g''$ should be discarded, as it never recommends Mozart. This leads to our first rule: if a hypothesis exhibits a high recall loss, it can be discarded. The second rule addresses precision loss, which is more challenging because it cannot be directly estimated from the data. To illustrate the second rule, imagine that in the training set, users tend to pick Bach pieces only 5% of the time. This suggests that $g'$ is over-recommending Bach pieces, and therefore, $g'$ might also be discarded based on its likely precision loss.

We formally capture this using a metric defined in the following. For any hypothesis $g$, let $U_i^g$ denote the uniform distribution over the output set $g(x_i)$ at $x_i$. Then, for any hypothesis $g$, we define a vector $v_g : \mathcal{H} \times \mathcal{H} \to [0, 1]$ by

$$v_g(g', g'') = \frac{1}{m} \sum_{i=1}^m U_i^g(g'(x_i) \setminus g''(x_i)) := \frac{1}{m} \sum_{i=1}^m \mathbb{P}_{v \sim U_i^g}(v \in g'(x_i) \setminus g''(x_i)).$$

Intuitively, $v_g(g', g'')$ represents the fraction of items inputs like that are recommended by $g'$ but not by $g''$ in the counterfactual scenario where $g$ is the target hypothesis. If $g$ is indeed the target hypothesis, then this quantity should be consistent with our training data, $v_g(g', g'') \approx v_{\widehat{g}}(g', g'')$, where $\widehat{g}$ is the observed (empirical) hypothesis; i.e., the hypothesis in which every $x_i$ is mapped to the random number $v_i$ which is observed in the training set. In the above example, $v_{g'}(g', g'')$ is 20% while $v_{\widehat{g}}(g', g'')$ is 5% and thus $g'$ is unlikely to be the target hypothesis.

We define a metric $d_{\mathcal{H}}$ between two hypotheses $g_1$ and $g_2$ by

$$d_{\mathcal{H}}(g_1, g_2) = \|v_{g_1} - v_{g_2}\|_\infty.$$

Surprisingly, we show that $d_{\mathcal{H}}(g^{\text{target}}, g)$ is a surrogate for the scalar loss, providing both lower and upper bounds on the scalar loss with a constant multiplicative factor:

$$d_{\mathcal{H}}(g^{\text{target}}, g) \leq 2\ell^{\text{scalar}}(g) \leq 4d_{\mathcal{H}}(g^{\text{target}}, g) + 2 \min_{g' \in \mathcal{H}} \ell^{\text{scalar}}(g').$$

A standard union bound argument yields that with probability at least $1 - \delta$,

$$d_{\mathcal{H}}(\widehat{g}, g^{\text{target}}) \leq O\left(\sqrt{\frac{\log|\mathcal{H}| + \log(1/\delta)}{m}}\right).$$

By triangle inequality, we have

$$d_{\mathcal{H}}(g^{\text{target}}, g) \leq d_{\mathcal{H}}(\widehat{g}, g) + O\left(\sqrt{\frac{\log |\mathcal{H}| + \log(1/\delta)}{m}}\right).$$

Then, we return a hypothesis $g^{\text{output}} \in \mathcal{H}$ such that

$$d_{\mathcal{H}}(\widehat{g}, g^{\text{output}}) = \min_{g \in \mathcal{H}} d_{\mathcal{H}}(\widehat{g}, g).$$

**The Hardness of No Knowledge of the Target Hypothesis's Output Size**  For any hypothesis $g$, it's precision loss at any input $x$ is $\frac{|g(x) \backslash g^{\text{target}}(x)|}{n_g(x)}$ and we only get a random label $v \sim \text{Unif}(g^{\text{target}}(x))$. If we are given the knowledge of the output size $n_{g^{\text{target}}}(x)$ of the target hypothesis, then we can obtain an unbiased estimate of the precision loss, i.e., $1 - \frac{n_{g^{\text{target}}}(x)}{n_g(x)} \cdot \mathbb{1}(v \in g(x))$. But the difficulty lies in that we don't know $n_{g^{\text{target}}}(x)$.

Consider the following example illustrated in Fig 3. For a given input $x$, there is a set of $n$ items equally divided into two sets $N_1(x)$ and $N_2(x)$. Consider two hypotheses—$g_1$ with output set $g_1(x) = N_1(x)$ and $g_2$ with output set $g_2(x) = N_1(x) \cup N_2(x)$ being all $n$ items.

In a world characterized by $\beta \in [\frac{1}{8}, \frac{2}{3}]$, the target hypothesis is generated in the following random way: Randomly select $\frac{3}{4} \cdot \beta n$ items from $N_1(x)$ and $\frac{1}{4} \cdot \beta n$ items from $N_2(x)$. No matter what $\beta$ is, w.p. $\frac{3}{4}$, $v$ is sampled uniformly at random from $N_1(x)$ and w.p. $\frac{1}{4}$, $v$ is sampled uniformly at random from $N_2(x)$. That is, every item in $N_1(x)$ has probability $\frac{3}{2n}$ of being sampled and every item in $N_2(x)$ has probability $\frac{1}{2n}$ of being sampled. Hence, if we have never seen the same input twice, we cannot distinguish between different $\beta$'s.

For any $g^{\text{target}}$ generated from the above process, the scalar loss of $g_1$ at $x$ is

$$\ell^{\text{scalar}}(g_1, x) = 1 - \left(\frac{|g^{\text{target}}(x) \cap g_1(x)|}{2|g_1(x)|} + \frac{|g^{\text{target}}(x) \cap g_1(x)|}{2|g^{\text{target}}(x)|}\right) = 1 - \left(\frac{3/4 \cdot \beta n}{n} + \frac{3/4 \cdot \beta n}{2\beta n}\right)$$
$$= \frac{5}{8} - \frac{3}{4}\beta,$$

and the scalar loss of $g_2$ at $x$ is

$$\ell^{\text{scalar}}(g_2, x) = 1 - \left(\frac{|g^{\text{target}}(x) \cap g_2(x)|}{2|g_2(x)|} + \frac{|g^{\text{target}}(x) \cap g_2(x)|}{2|g^{\text{target}}(x)|}\right) = 1 - \left(\frac{\beta n}{2n} + \frac{\beta n}{2\beta n}\right) = \frac{1}{2} - \frac{1}{2}\beta.$$

When $\beta$ is large, $g_1$ has a smaller loss; when $\beta$ is small, $g_2$ has a smaller loss. With a huge input space $\mathcal{X}$ and the distribution $\mathcal{D}$ being uniform over such a huge space $\mathcal{X}$, it is almost sure that we will never observe the same input twice and, therefore, cannot distinguish between different $\beta$ values. Consequently, it's impossible to determine which of the two hypotheses has a smaller loss. We show that, in this example, even if algorithms are allowed to be randomized and improper, it still impossible to compete with the best hypothesis in the hypothesis class.

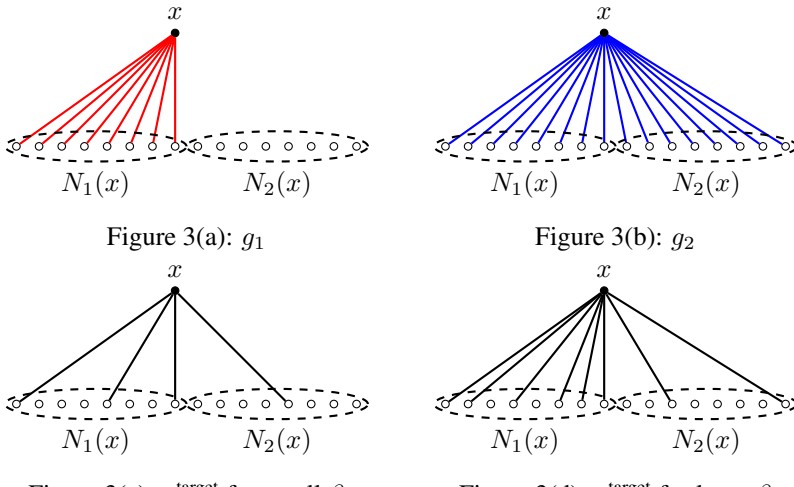

Figure 3(a): $g_1$  Figure 3(b): $g_2$

Figure 3(c): $g^{\text{target}}$ for small $\beta$   Figure 3(d): $g^{\text{target}}$ for large $\beta$

Figure 3: Illustration of $g_1$, $g_2$ and randomly generated $g^{\text{target}}$.

## B   Algorithms and Proofs

**Notations**   Let $\ell^{\text{precision}}(g,x)$ and $\ell^{\text{recall}}(g,x)$ denote the precision loss and recall loss of hypothesis $g$ at input $x$:

$$\ell^{\text{precision}}(g,x) = \frac{|g(x) \setminus g^{\text{target}}(x)|}{n_g(x)},$$

$$\ell^{\text{recall}}(g,x) = \frac{|g^{\text{target}}(x) \setminus g(x)|}{n_{g^{\text{target}}}(x)}.$$

Then the empirical precision and recall losses are $\widehat{\ell}^{\text{precision}}(g) = \frac{1}{m}\sum \ell^{\text{precision}}(g,x_i)$ and $\widehat{\ell}^{\text{recall}}(g) = \frac{1}{m}\sum \ell^{\text{recall}}(g,x_i)$. Let $a \wedge b := \min(a,b)$.

### B.1   Maximum Likelihood Method in the Realizable Case

In the realizable setting, the target hypothesis $g^{\text{target}}$ is in the hypothesis class. Given the IID training data $(x_1,v_1),\ldots,(x_m,v_m)$, the maximum likelihood method returns the hypothesis

$$g^{\text{output}} = \arg\max_{g \in \mathcal{H}} \prod_{i=1}^{m} \frac{\mathbb{1}(v_i \in g(x_i))}{n_g(x_i)}.$$

In other words, $g^{\text{output}}$ is a hypothesis in $\mathcal{H}$ satisfying

- consistency: $\sum_{i=1}^{m} \mathbb{1}(v_i \notin g^{\text{output}}(x_i)) = 0$;
- regulation: among all consistent hypotheses, the output hypothesis satisfies that $g^{\text{output}} = \arg\min_{g:g \text{ is consistent}} \sum_{i=1}^{m} \log(n_g(x_i))$.[7]

**Theorem 1.** *In the realizable setting, there exist algorithms such that given an IID training set of size $m \geq O(\frac{\log(|\mathcal{H}|/\delta)}{\varepsilon})$, with probability at least $1 - \delta$, the output hypothesis $g^{\text{output}}$ satisfies*

$$\ell^{\text{recall}}(g^{\text{output}}) \leq \varepsilon, \quad \ell^{\text{precision}}(g^{\text{output}}) \leq \varepsilon.$$

*Proof of Theorem 1.* It suffices to prove that for any fixed $(x_1,\ldots,x_m)$, when $m \geq \frac{12\log(4|\mathcal{H}|/\delta)}{\varepsilon}$, w.p. at least $1 - \delta/2$ over $v_{1:m}$, the empirical values of recall and precision are small, $\widehat{\ell}^{\text{recall}}(g^{\text{output}}) \leq \varepsilon/2$ and $\widehat{\ell}^{\text{precision}}(g^{\text{output}}) \leq \varepsilon/2$. Then we can show $\ell^{\text{recall}}(g^{\text{output}}) \leq \varepsilon$ and $\ell^{\text{precision}}(g^{\text{output}}) \leq \varepsilon$ by applying empirical Bernstein bounds to both precision and recall losses. Now we prove this statement.

---

[7]the base of $\log$ in this work is 2.

Bounding recall loss is easy as $\sum_{i=1}^{m} \mathbb{1}(v_i \notin g(x_i))$ is an unbiased estimate of recall loss. With probability $1 - \delta/4$ over the randomness of $v_i \sim \text{Unif}(g^{\text{target}}(x_i))$ for all $i \in [m]$, the empirical loss for recall is

$$\widehat{\ell}^{\text{recall}}(g) = \frac{1}{m} \sum_{i=1}^{m} \frac{|g^{\text{target}}(x_i) \setminus g(x_i)|}{n_{g^{\text{target}}}(x_i)} \leq \sum_{i=1}^{m} \mathbb{1}(v_i \notin g(x_i)) + \varepsilon/6 = \varepsilon/6 \,, \tag{3}$$

for all consistent $g$ with $\sum_{i=1}^{m} \mathbb{1}(v_i \notin g(x_i)) = 0$. Hence, w.p. $1 - \delta/4$, $\widehat{\ell}^{\text{recall}}(g^{\text{output}}) \leq \varepsilon/6$.

Bounding precision loss is more challenging. Let $A_g = \{i \in [m] | n_{g^{\text{target}}}(x_i) \leq 2n_g(x_i)\}$. Then we can decompose the empirical precision loss as

$$\widehat{\ell}^{\text{precision}}(g)$$

$$= \frac{1}{m} \sum_{i=1}^{m} \frac{|g(x_i) \setminus g^{\text{target}}(x_i)|}{n_g(x_i)}$$

$$\leq \frac{1}{m} \sum_{i \in A_g} \frac{|g(x_i) \setminus g^{\text{target}}(x_i)|}{n_g(x_i)} + \frac{1}{m} \sum_{i=1}^{m} \mathbb{1}(i \notin A_g)$$

$$\leq \frac{1}{m} \sum_{i \in A_g} \min\left( \frac{2|g(x_i) \setminus g^{\text{target}}(x_i)|}{n_{g^{\text{target}}}(x_i)}, 1 \right) + \frac{2}{m} \sum_{i \notin A_g} \frac{|g^{\text{target}}(x_i) \setminus g(x_i)|}{n_{g^{\text{target}}}(x_i)}$$

$$= \frac{2}{m} \sum_{i \in A_g} \min\left( \frac{n_g(x_i) + |g^{\text{target}}(x_i) \setminus g(x_i)| - g^{\text{target}}(x_i)}{n_{g^{\text{target}}}(x_i)}, \frac{1}{2} \right) + \frac{2}{m} \sum_{i \notin A_g} \frac{|g^{\text{target}}(x_i) \setminus g(x_i)|}{n_{g^{\text{target}}}(x_i)}$$

$$\leq \frac{2}{m} \sum_{i \in A_g} \min\left( \frac{n_g(x_i)}{n_{g^{\text{target}}}(x_i)} - 1, \frac{1}{2} \right) + \frac{2}{m} \sum_{i \in A_g} \frac{|g^{\text{target}}(x_i) \setminus g(x_i)|}{n_{g^{\text{target}}}(x_i)} + \frac{2}{m} \sum_{i \notin A_g} \frac{|g^{\text{target}}(x_i) \setminus g(x_i)|}{n_{g^{\text{target}}}(x_i)}$$

$$\leq \frac{2}{m} \sum_{i \in A_g} \min\left( \frac{n_g(x_i)}{n_{g^{\text{target}}}(x_i)} - 1, \frac{1}{2} \right) + 2\widehat{\ell}^{\text{recall}}(g) \,.$$

The second term is the empirical loss for recall, which is upper bounded by Eq (3).

For the first term,

$$\frac{1}{m} \sum_{i \in A_g} \min\left( \frac{n_g(x_i)}{n_{g^{\text{target}}}(x_i)} - 1, \frac{1}{2} \right)$$

$$= \frac{1}{m} \sum_{i: \frac{n_g(x_i)}{n_{g^{\text{target}}}(x_i)} \geq \frac{1}{2}} \min\left( \frac{n_g(x_i)}{n_{g^{\text{target}}}(x_i)} - 1, \frac{1}{2} \right)$$

$$\leq \frac{1}{m} \sum_{i: \frac{n_g(x_i)}{n_{g^{\text{target}}}(x_i)} \geq 1} \min\left( \frac{n_g(x_i)}{n_{g^{\text{target}}}(x_i)} - 1, \frac{1}{2} \right)$$

$$\leq \sum_{i: \frac{n_g(x_i)}{n_{g^{\text{target}}}(x_i)} \geq 1} \log \frac{n_g(x_i)}{n_{g^{\text{target}}}(x_i)} \,,$$

where the last inequality adopts the following fact: for all $z \geq 1$, $\min(z - 1, \frac{1}{2}) \leq \log z$. On the other hand, we have

$$\frac{1}{m} \sum_{i: \frac{n_g(x_i)}{n_{g^{\text{target}}}(x_i)} \geq 1} \log \frac{n_g(x_i)}{n_{g^{\text{target}}}(x_i)} \leq \frac{1}{m} \sum_{i: \frac{n_g(x_i)}{n_{g^{\text{target}}}(x_i)} < 1} \log \frac{n_{g^{\text{target}}}(x_i)}{n_g(x_i)}$$

when $g$ satisfies $\sum_{i=1}^{m} \log(n_g(x_i)) \leq \sum_{i=1}^{m} \log(n_{g^{\text{target}}}(x_i))$. Hence, for any hypothesis $g$ with $\widehat{\ell}^{\text{precision}}(g) > \frac{\varepsilon}{2}$ and $\widehat{\ell}^{\text{recall}}(g) \leq \frac{\varepsilon}{6}$, we have

$$\sum_{i: \frac{n_g(x_i)}{n_{g^{\text{target}}}(x_i)} < 1} \log \frac{n_{g^{\text{target}}}(x_i)}{n_g(x_i)} \geq \frac{m}{2} (\widehat{\ell}^{\text{precision}}(g) - 2\widehat{\ell}^{\text{recall}}(g)) > m \cdot \varepsilon/12 \geq \log(4|\mathcal{H}|/\delta) \,.$$

The probability of outputting such a hypothesis $g$ is

$$\mathbb{P}_{v_{1:m}}\left(g^{\text{output}} = g\right) \leq \mathbb{P}_{v_{1:m}}(g \text{ is consistent}) \leq \prod_{i:\frac{n_g(x_i)}{n_{g^{\text{target}}}(x_i)}<1} \frac{n_g(x_i)}{n_{g^{\text{target}}}(x_i)} < \frac{\delta}{4|\mathcal{H}|}\,.$$

Hence, with probability at least $1 - \delta/2$ over $v_{1:m}$, $g^{\text{output}}$ will satisfy

$$\widehat{\ell}^{\text{recall}}(g^{\text{output}}) \leq \varepsilon/6\,, \widehat{\ell}^{\text{precision}}(g^{\text{output}}) \leq \varepsilon/2\,.$$

Then we are done with the proof. $\qquad\square$

## B.2 Modified Maximum Likelihood Method in the Agnostic Case

In the agnostic setting, we shift our goal from finding a hypothesis with nearly zero precision and recall losses to determining, given any $p, r \in [0, 1]$, the minimum precision loss $p'$ such that $(p, r) \Rightarrow (p', r)$.

In fact, we can show something stronger: we do not need to know $p$. Specifically, given any $r \in [0, 1]$, let $p = \min_{g:\ell^{\text{recall}}(g)\leq r} \ell^{\text{precision}}(g)$ be the optimal precision loss among all hypotheses with recall loss at most $r$. What is the smallest $p'$ such that we achieve $\ell^{\text{recall}}(g^{\text{output}}) \leq r$ and $\ell^{\text{precision}}(g^{\text{output}}) \leq p'$?

**Theorem 3.** *There exists a class $\mathcal{H} = \{g_1, g_2\}$ of two hypotheses, such that for any (possibly randomized improper) algorithm given the knowledge of $(p, r) = (\frac{7}{16}, \frac{1}{4})$, there exists a target hypothesis $g^{\text{target}}$ with bounded output set size and a data distribution $\mathcal{D}$ for which there exists a hypothesis $g^\dagger \in \mathcal{H}$ with $\ell^{\text{recall}}(g^\dagger) = \frac{1}{4}$ and $\ell^{\text{precision}}(g^\dagger) = \frac{7}{16}$ s.t. for any sample size $m > 0$, with probability $1$ over the training set, the expected (over the randomness of the algorithm) precision and recall losses of the output $g^{\text{output}}$ satisfy*

$$\mathbb{E}\left[\ell^{\text{recall}}(g^{\text{output}})\right] + \frac{12}{5}\mathbb{E}\left[\ell^{\text{precision}}(g^{\text{output}})\right] \geq \frac{7}{5}\,.$$

Recall that in the realizable case, the maximum likelihood method selects the consistent hypothesis with the smallest empirical log output size, i.e., $g^{\text{output}} = \arg\min_{g:g \text{ is consistent}} \sum_{i=1}^{m} \log(n_g(x_i))$. Basically, the consistency guarantees small recall loss and the empirical log output size is used as a regulation term to bound precision loss. In the agnostic setting, the maximum likelihood method fails as there may be no consistent hypothesis. We present a modified version of the maximum likelihood method, which still has the "consistency" component and adopts log output size as a regulation term. The algorithm operates as follows.

---

**Algorithm:**

1. **Finding a set of plausible hypotheses which make at most $m \cdot (r + \varepsilon)$ mistakes**: For any hypothesis $g$, let $I_g = \{i | v_i \notin g(x_i)\}$ denote the indices of training points that the hypothesis $g$ is inconsistent with. Then let $\widehat{\mathcal{H}} = \{g \in \mathcal{H} | |I_g| \leq m \cdot (r + \varepsilon)\}$ be the set of hypotheses making at most $m \cdot (r + \varepsilon)$ mistakes.

2. **Returning the hypothesis with low empirical log truncated output size**: Return

$$g^{\text{output}} = \arg\min_{g' \in \widehat{\mathcal{H}}} \max_{g \in \widehat{\mathcal{H}}} \frac{1}{m} \sum_{i \in [m]} \log \frac{n_{g'}(x_i) \wedge 4n_g(x_i)}{n_g(x_i) \wedge 4n_{g'}(x_i)}\,. \tag{4}$$

---

**Theorem 7.** *Given any $r \in [0, 1]$ and $m \geq O(\frac{\log(|\mathcal{H}|)+\log(1/\delta)}{\varepsilon^2})$ IID training data, the modified maximum likelihood method can return a hypothesis satisfying*

$$\ell^{\text{recall}}(g^{\text{output}}) \leq r + \varepsilon, \quad \ell^{\text{precision}}(g^{\text{output}}) \leq 28r + 15p + \varepsilon\,,$$

*where $p = \min_{g:\ell^{\text{recall}}(g)\leq r} \ell^{\text{precision}}(g)$.*

*Proof Sketch.* Let $g^\dagger$ to be the hypothesis with recall loss at most $r$ and precision loss $p$. Let $\bar{r} = r + 2\varepsilon$ and $\bar{p} = p + 2\varepsilon$. Similar to the realizable setting, we build connection between precision

and recall with empirical log output size (Lemma 2), i.e., for any constant $c \in (0, 1]$,

$$\widehat{\ell}^{\text{precision}}(g) \le \frac{1+c}{m} \sum_{i \in [m]: \frac{n_g(x_i)}{n_{g^{\text{target}}}(x_i)} \ge 1} \log \frac{n_g(x_i) \wedge 2n_{g^{\text{target}}}(x_i)}{n_{g^{\text{target}}}(x_i)} + \frac{1+c}{c} \widehat{\ell}^{\text{recall}}(g) \,.$$

It suffices to upper bound $\sum_{i \in [m]: \frac{n_g(x_i)}{n_{g^{\text{target}}}(x_i)} \ge 1} \log \frac{n_g(x_i) \wedge 2n_{g^{\text{target}}}(x_i)}{n_{g^{\text{target}}}(x_i)}$. We first decompose it into

$$\sum_{i \in [m]: \frac{n_g(x_i)}{n_{g^{\text{target}}}(x_i)} \ge 1} \log \frac{n_g(x_i) \wedge 2n_{g^{\text{target}}}(x_i)}{n_{g^{\text{target}}}(x_i)}$$

$$= \underbrace{\sum_{i \in B} \log \frac{n_g(x_i) \wedge 2n_{g^{\text{target}}}(x_i)}{n_{g^{\text{target}}}(x_i)}}_{(a)} - \underbrace{\sum_{i \in B: \frac{n_g(x_i)}{n_{g^{\text{target}}}(x_i)} < 1} \log \frac{n_g(x_i) \wedge 2n_{g^{\text{target}}}(x_i)}{n_{g^{\text{target}}}(x_i)}}_{(b)} \,,$$

where $B = \{i \mid \frac{n_g(x_i)}{n_{g^{\text{target}}}(x_i)} \ge \frac{1}{2}\}$. The term (b) is lower bounded by the recall loss in Lemma 4. Intuitively, if $\frac{n_g(x_i) \wedge 2n_{g^{\text{target}}}(x_i)}{n_{g^{\text{target}}}(x_i)}$ is small, the recall loss must be large while any hypothesis in $\widehat{\mathcal{H}}$ has a small empirical recall loss.

For term (a), since $\widehat{\ell}^{\text{recall}}(g^\dagger) \le \bar{r}$ and $\widehat{\ell}^{\text{precision}}(g^\dagger) \le \bar{p}$, at most training points, $\frac{n_{g^\dagger}(x_i)}{n_{g^{\text{target}}}(x_i)}$ is in $[\frac{1}{2}, 2]$ (if it's too large at $x_i$, precision loss is large; if it's too small, recall loss is large). Also, for any hypothesis in $\widehat{\mathcal{H}}$, the empirical recall loss is small and thus, at most training points, $\frac{n_g(x_i)}{n_{g^{\text{target}}}(x_i)} \ge \frac{1}{2}$. At these training points satisfying $\frac{n_{g^\dagger}(x_i)}{n_{g^{\text{target}}}(x_i)}$ is in $[\frac{1}{2}, 2]$ and $\frac{n_{g^{\text{output}}}(x_i)}{n_{g^{\text{target}}}(x_i)} \ge \frac{1}{2}$, we have

$$\log \frac{n_g(x_i) \wedge 2n_{g^{\text{target}}}(x_i)}{n_{g^{\text{target}}}(x_i)} \le \log \frac{n_g(x_i) \wedge 4n_{g^\dagger}(x_i)}{n_{g^{\text{target}}}(x_i)}$$

$$= \log \frac{n_g(x_i) \wedge 4n_{g^\dagger}(x_i)}{n_{g^\dagger}(x_i) \wedge 4n_g(x_i)} + \log \frac{n_{g^\dagger}(x_i) \wedge 4n_g(x_i)}{n_{g^{\text{target}}}(x_i)}$$

$$\le \log \frac{n_g(x_i) \wedge 4n_{g^\dagger}(x_i)}{n_{g^\dagger}(x_i) \wedge 4n_g(x_i)} + \log \frac{n_{g^\dagger}(x_i)}{n_{g^{\text{target}}}(x_i)} \,.$$

The first term is bounded due to our algorithm while the second term is bounded by the empirical precision (in Lemma 5). Intuitively, if $\frac{n_{g^\dagger}(x_i)}{n_{g^{\text{target}}}(x_i)}$ is large, the empirical precision of $g^\dagger$ is large. $\quad\square$

### B.2.1 Proof of Theorem 3

*Proof of Theorem 3.* Consider the size of the input space $\mathcal{X}$ being infinite and $\mathcal{D}$ being the uniform distribution over it. For each input $x$, there is an individual set of 12 items, denoted as $N(x) = \{v_{x,1}, v_{x,2}, \ldots, v_{x,12}\}$. The hypothesis $g_1$ outputs the first 8 items $g_1(x) = \{v_{x,1}, v_{x,2}, \ldots, v_{x,8}\}$ and the hypothesis $g_2$ outputs the last 8 items $g_2(x) = \{v_{x,5}, v_{x,6}, \ldots, v_{x,12}\}$. There are two worlds in which the target hypothesis $g^{\text{target}}$ is generated differently:

- **World I:** W.p. $\frac{1}{2}$, $g^{\text{target}}(x) = g_1(x)$; w.p. $\frac{1}{2}$, $g^{\text{target}}(x) = \{u_1, u_2\}$ where $u_1$ is sampled uniformly from $\{v_{x,5}, v_{x,6}, \ldots, v_{x,8}\}$ and $u_2$ is sampled uniformly from $\{v_{x,9}, v_{x,10}, \ldots, v_{x,12}\}$.

- **World II:** W.p. $\frac{1}{2}$, $g^{\text{target}}(x) = g_2(x)$; w.p. $\frac{1}{2}$, $g^{\text{target}}(x) = \{u_1, u_2\}$ where $u_1$ is sampled uniformly from $\{v_{x,5}, v_{x,6}, \ldots, v_{x,8}\}$ and $u_2$ is sampled uniformly from $\{v_{x,1}, v_{x,2}, \ldots, v_{x,4}\}$.

These two worlds are symmetric. In world I,

$$\text{recall}(g_1) = \frac{1}{2} \cdot 1 + \frac{1}{2} \cdot \frac{1}{2} = \frac{3}{4}, \quad \text{precision}(g_1) = \frac{1}{2} \cdot 1 + \frac{1}{2} \cdot \frac{1}{8} = \frac{9}{16} \,,$$

$$\text{recall}(g_2) = \frac{1}{2} \cdot \frac{1}{2} + \frac{1}{2} \cdot 1 = \frac{3}{4}, \quad \text{precision}(g_2) = \frac{1}{2} \cdot \frac{1}{2} + \frac{1}{2} \cdot \frac{1}{4} = \frac{3}{8} \,.$$

Both $g_1$ and $g_2$ have the same recall and $g_1$ has a better precision in the world I. In world II, $g_1$ and $g_2$ switch their losses. Hence, in either world, we have

$$\min_{g \in \mathcal{H}} \ell^{\text{recall}}(g) = \frac{1}{4}, \quad \min_{g \in \mathcal{H}} \ell^{\text{precision}}(g) = \frac{7}{16}.$$

In both worlds, the distribution of $v$ is identical, i.e., w.p. $\frac{1}{2}$, $v$ is sampled from $\text{Unif}(g_1(x))$ and w.p. $\frac{1}{2}$, $v$ is sampled from $\text{Unif}(g_2(x))$. Hence, if no $x$ has been sampled twice, no algorithm can distinguish the two worlds. Since the input space is huge, we will not sample the same input twice almost surely. For any output hypothesis $g^{\text{output}}$, we let

$$n_1 = |g^{\text{output}}(x) \cap \{v_{x,1}, v_{x,2}, \dots, v_{x,4}\}| \in \{0, 1, \dots, 4\},$$
$$n_2 = |g^{\text{output}}(x) \cap \{v_{x,5}, v_{x,6}, \dots, v_{x,8}\}| \in \{0, 1, \dots, 4\},$$
$$n_3 = |g^{\text{output}}(x) \cap \{v_{x,9}, v_{x,10}, \dots, v_{x,12}\}| \in \{0, 1, \dots, 4\}.$$

Then in world I, the expected recall and precision of $g^{\text{output}}$ is

$$\mathbb{E}_{g^{\text{target}}} \left[\text{recall}(g^{\text{output}}, x)\right] = \frac{1}{2} \left(\frac{n_1 + n_2}{8} + \frac{n_2 + n_3}{8}\right) = \frac{n_1 + 2n_2 + n_3}{16},$$
$$\mathbb{E}_{g^{\text{target}}} \left[\text{precision}(g^{\text{output}}, x)\right] = \frac{1}{2} \left(\frac{n_1 + n_2}{n_1 + n_2 + n_3} + \frac{n_2 + n_3}{4(n_1 + n_2 + n_3)}\right) = \frac{4n_1 + 5n_2 + n_3}{8(n_1 + n_2 + n_3)}.$$

Then suppose we randomly choose one of the two world. By taking expectation over the world, $x$, and the randomness of the algorithm, we have

$$\mathbb{E}\left[(\text{recall}(g^{\text{output}}), \text{precision}(g^{\text{output}}))\right] = \mathbb{E}\left[\left(\frac{n_1 + 2n_2 + n_3}{16}, \frac{5(n_1 + 2n_2 + n_3)}{16(n_1 + n_2 + n_3)}\right)\right]$$
$$= \mathbb{E}\left[\left(\frac{n_1 + 2n_2 + n_3}{16}, \frac{5}{16} + \frac{5}{16} \cdot \frac{n_2}{n_1 + n_2 + n_3}\right)\right]$$

Let's denote by $r(n_1, n_2, n_3) = \frac{n_1 + 2n_2 + n_3}{16}$ and $p(n_1, n_2, n_3) = \frac{5}{16} + \frac{5}{16} \cdot \frac{n_2}{n_1 + n_2 + n_3}$. Then for any $(n_1, n_2, n_3) \in \{0, \dots, 4\}^3$, we have

$$r(n_1, n_2, n_3) + \frac{12}{5} p(n_1, n_2, n_3)$$
$$= \frac{n_1 + 2n_2 + n_3}{16} + \frac{3}{4} + \frac{3}{4} \cdot \frac{n_2}{n_1 + n_2 + n_3}$$
$$\leq \frac{n_1 + 8 + n_3}{16} + \frac{3}{4} + \frac{3}{n_1 + 4 + n_3} \qquad \text{(maximized at } n_2 = 4\text{)}$$
$$\leq 2. \qquad \text{(maximized at } n_1 + n_3 = 0 \text{ or } 8\text{)}$$

Hence, we have

$$\mathbb{E}\left[\text{recall}(g^{\text{output}})\right] + \frac{12}{5} \mathbb{E}\left[\text{precision}(g^{\text{output}})\right] \leq 2.$$

This is equivalent to

$$\mathbb{E}\left[\ell^{\text{recall}}(g^{\text{output}})\right] + \frac{12}{5} \mathbb{E}\left[\ell^{\text{precision}}(g^{\text{output}})\right] \geq \frac{7}{5}.$$

$\square$

### B.2.2  Proof of Theorem 7

*Proof of Theorem 7.* Let $\Delta = \sqrt{\frac{\log(|\mathcal{H}|/\delta)}{m}}$, $\bar{r} = r + 2\Delta$ and $\bar{p} = p + 2\Delta$. Let $g^{\dagger}$ denote the hypothesis with precision and recall losses $(p, r)$. We know that with probability at least $1 - \delta$, all hypotheses in $\widehat{\mathcal{H}}$ have empirical recall loss no greater than $\bar{r}$ and $g^{\dagger}$ also has $\widehat{\ell}^{\text{precision}}(g^{\dagger}) \leq \bar{p}$. Then the proof is divided into two parts:

(i) hypothesis $g^\dagger$ satisfies that

$$\max_{g \in \widehat{\mathcal{H}}} \frac{1}{m} \sum_{i \in [m]} \log \frac{n_{g^\dagger}(x_i) \wedge 4n_g(x_i)}{n_g(x_i) \wedge 4n_{g^\dagger}(x_i)} \leq 6\overline{r} + 4\overline{p} + \frac{2}{m} \,.$$

Therefore, we have $\max_{g \in \widehat{\mathcal{H}}} \frac{1}{m} \sum_{i \in [m]} \log \frac{n_{g^{\text{output}}}(x_i) \wedge 4n_g(x_i)}{n_g(x_i) \wedge 4n_{g^{\text{output}}}(x_i)} \leq 6\overline{r} + 4\overline{p} + \frac{2}{m}$.

(ii) any hypothesis $g'$ satisfying $\max_{g \in \widehat{\mathcal{H}}} \sum_{i \in [m]} \log \frac{n_{g'}(x_i) \wedge 4n_g(x_i)}{n_g(x_i) \wedge 4n_{g'}(x_i)} \leq 6m\overline{r} + 4m\overline{p} + 2$ has $\widehat{\ell}^{\text{precision}}(g') \leq 28r + 15p + o(1)$. Hence, $g^{\text{output}}$ can achieve good precision.

We prove part (i) by Lemma 1 and part (ii) by combining Lemma 2 and 3. $\qquad\square$

**Lemma 1.** *For any $x_1, \ldots, x_m$ and hypothesis $g$ with $\widehat{\ell}^{recall}(g) \leq \overline{r}$, we have*

$$\frac{1}{m} \sum_{i \in [m]} \log \frac{n_{g^\dagger}(x_i) \wedge 4n_g(x_i)}{n_g(x_i) \wedge 4n_{g^\dagger}(x_i)} \leq 6\overline{r} + 4\overline{p} + \frac{2}{m} \,.$$

*Proof.* Let $E = \{i | \frac{n_g(x_i)}{n_{g^{\text{target}}}(x_i)} \geq \frac{1}{2}, \frac{1}{2} \leq \frac{n_{g^\dagger}(x_i)}{n_{g^{\text{target}}}(x_i)} \leq 2\}$. According to Lemma 6, we know $|\neg E| \leq 4m\overline{r} + 2m\overline{p}$.

$$
\begin{aligned}
\sum_{i \in [m]} \log \frac{n_{g^\dagger}(x_i) \wedge 4n_g(x_i)}{n_g(x_i) \wedge 4n_{g^\dagger}(x_i)} &\leq \sum_{i \in E} \log \frac{n_{g^\dagger}(x_i) \wedge 4n_g(x_i)}{n_g(x_i) \wedge 4n_{g^\dagger}(x_i)} + |\neg E| \\
&\leq \sum_{i \in E} \log \frac{n_{g^\dagger}(x_i)}{n_g(x_i) \wedge n_{g^{\text{target}}}(x_i)} + 4m\overline{r} + 2m\overline{p} \\
&= \sum_{i \in E} \log \frac{n_{g^\dagger}(x_i)}{n_{g^{\text{target}}}(x_i)} - \sum_{i \in E} \log \frac{n_g(x_i) \wedge n_{g^{\text{target}}}(x_i)}{n_{g^{\text{target}}}(x_i)} + 4m\overline{r} + 2m\overline{p} \\
&\leq 6m\overline{r} + 4m\overline{p} + 2 \,. \qquad \text{(Applying Lemmas 4 and 5)}
\end{aligned}
$$

$\qquad\square$

**Lemma 2.** *For any $x_1, \ldots, x_m$, any hypothesis $g$, and any positive constant $c \in (0, 1]$, the empirical loss for precision $\widehat{\ell}^{precision}(g)$ satisfies*

$$\widehat{\ell}^{precision}(g) \leq \frac{1+c}{m} \sum_{i \in [m]: \frac{n_g(x_i)}{n_{g^{target}}(x_i)} \geq 1} \log \frac{n_g(x_i) \wedge 2n_{g^{target}}(x_i)}{n_{g^{target}}(x_i)} + \frac{1+c}{c} \widehat{\ell}^{recall}(g) \,.$$

*Proof of Lemma 2.* Let $A_g = \{i \in [m] | n_{g^{\text{target}}}(x_i) \leq (1+c)n_g(x_i)\}$. Then we have

$$\widehat{\ell}^{\text{precision}}(g)$$

$$= \frac{1}{m} \sum_{i=1}^{m} \frac{|g(x_i) \setminus g^{\text{target}}(x_i)|}{n_g(x_i)}$$

$$\leq \frac{1}{m} \sum_{i \in A_g} \frac{|g(x_i) \setminus g^{\text{target}}(x_i)|}{n_g(x_i)} + \frac{1}{m} \sum_{i=1}^{m} \mathbb{1}(i \notin A_g)$$

$$\leq \frac{1}{m} \sum_{i \in A_g} \min\left(\frac{(1+c)|g(x_i) \setminus g^{\text{target}}(x_i)|}{n_{g^{\text{target}}}(x_i)}, 1\right) + \frac{1+c}{c \cdot m} \sum_{i \notin A_g} \frac{|g^{\text{target}}(x_i) \setminus g(x_i)|}{n_{g^{\text{target}}}(x_i)}$$

$$\leq \frac{1+c}{m} \sum_{i \in A_g} \min\left(\frac{n_g(x_i) + |g^{\text{target}}(x_i) \setminus g(x_i)| - n_{g^{\text{target}}}(x_i)}{n_{g^{\text{target}}}(x_i)}, 1\right) + \frac{1+c}{c \cdot m} \sum_{i \notin A_g} \frac{|g^{\text{target}}(x_i) \setminus g(x_i)|}{n_{g^{\text{target}}}(x_i)}$$

$$\leq \frac{1+c}{m} \sum_{i \in A_g} \min\left(\frac{n_g(x_i)}{n_{g^{\text{target}}}(x_i)} - 1, 1\right) + \frac{1+c}{m} \sum_{i \in A_g} \frac{|g^{\text{target}}(x_i) \setminus g(x_i)|}{n_{g^{\text{target}}}(x_i)} + \frac{1+c}{c \cdot m} \sum_{i \notin A_g} \frac{|g^{\text{target}}(x_i) \setminus g(x_i)|}{n_{g^{\text{target}}}(x_i)}$$

$$\leq \frac{1+c}{m} \sum_{i \in A_g} \min\left(\frac{n_g(x_i)}{n_{g^{\text{target}}}(x_i)} - 1, 1\right) + \frac{1+c}{c} \widehat{\ell}^{\text{recall}}(g).$$

Now we upper bound the first term.

$$\sum_{i \in A_g} \min\left(\frac{n_g(x_i)}{n_{g^{\text{target}}}(x_i)} - 1, 1\right)$$

$$= \sum_{i \in [m]: \frac{n_g(x_i)}{n_{g^{\text{target}}}(x_i)} \geq \frac{1}{1+c}} \min\left(\frac{n_g(x_i)}{n_{g^{\text{target}}}(x_i)} - 1, 1\right)$$

$$\leq \sum_{i \in [m]: \frac{n_g(x_i)}{n_{g^{\text{target}}}(x_i)} \geq 1} \min\left(\frac{n_g(x_i)}{n_{g^{\text{target}}}(x_i)} - 1, 1\right)$$

$$\leq \sum_{i \in [m]: \frac{n_g(x_i)}{n_{g^{\text{target}}}(x_i)} \geq 1} \min\left(\log \frac{n_g(x_i)}{n_{g^{\text{target}}}(x_i)}, 1\right)$$

$$= \sum_{i \in [m]: \frac{n_g(x_i)}{n_{g^{\text{target}}}(x_i)} \geq 1} \log \frac{n_g(x_i) \wedge 2n_{g^{\text{target}}}(x_i)}{n_{g^{\text{target}}}(x_i)},$$

where the last inequality adopts the fact: for all $z \geq 1$, $\min(z - 1, 1) \leq \log z$. $\qquad\square$

**Lemma 3.** *For any $g$ satisfying $\widehat{\ell}^{recall}(g) \leq \bar{r}$ and $\frac{1}{m} \sum_{i \in [m]} \log \frac{n_g(x_i) \wedge 4n_{g^\dagger}(x_i)}{n_{g^\dagger}(x_i) \wedge 4n_g(x_i)} \leq 6\bar{r} + 4\bar{p} + \frac{2}{m}$,
we have $\frac{1}{m} \sum_{i \in [m]: \frac{n_g(x_i)}{n_{g^{target}}(x_i)} \geq 1} \log \frac{n_g(x_i) \wedge 2n_{g^{target}}(x_i)}{n_{g^{target}}(x_i)} \leq 18\bar{r} + 12\bar{p} + \frac{4}{m}.$*

*Proof of Lemma 3.* Let $B = \{i | \frac{n_g(x_i)}{n_{g^{\text{target}}}(x_i)} \geq \frac{1}{2}\}$ and $C = \{i | \frac{1}{2} \leq \frac{n_{g^\dagger}(x_i)}{n_{g^{\text{target}}}(x_i)} \leq 2\}$. For any $i \in B$, the value of $\log \frac{n_g(x_i) \wedge 2n_{g^{\text{target}}}(x_i)}{n_{g^{\text{target}}}(x_i)}$ is in $[-1, 1]$. Then we have

$$\sum_{i \in [m]: \frac{n_g(x_i)}{n_{g^{\text{target}}}(x_i)} \geq 1} \log \frac{n_g(x_i) \wedge 2n_{g^{\text{target}}}(x_i)}{n_{g^{\text{target}}}(x_i)}$$

$$= \sum_{i \in B} \log \frac{n_g(x_i) \wedge 2n_{g^{\text{target}}}(x_i)}{n_{g^{\text{target}}}(x_i)} - \sum_{i \in B: \frac{n_g(x_i)}{n_{g^{\text{target}}}(x_i)} < 1} \log \frac{n_g(x_i) \wedge 2n_{g^{\text{target}}}(x_i)}{n_{g^{\text{target}}}(x_i)}$$

$$\leq \sum_{i \in C \cap B} \log \frac{n_g(x_i) \wedge 2n_{g^{\text{target}}}(x_i)}{n_{g^{\text{target}}}(x_i)} + |\neg C| - \sum_{i \in B} \log \frac{n_g(x_i) \wedge n_{g^{\text{target}}}(x_i)}{n_{g^{\text{target}}}(x_i)}$$

$$\leq \sum_{i \in C \cap B} \log \frac{n_g(x_i) \wedge 4n_{g^\dagger}(x_i)}{n_{g^{\text{target}}}(x_i)} + |\neg C| - \sum_{i \in B} \log \frac{n_g(x_i) \wedge n_{g^{\text{target}}}(x_i)}{n_{g^{\text{target}}}(x_i)}$$

$$= \sum_{i \in C \cap B} \log \frac{n_g(x_i) \wedge 4n_{g^\dagger}(x_i)}{n_{g^\dagger}(x_i) \wedge 4n_g(x_i)} + \sum_{i \in C \cap B} \log \frac{n_{g^\dagger}(x_i) \wedge 4n_g(x_i)}{n_{g^{\text{target}}}(x_i)} + |\neg C|$$

$$- \sum_{i \in B} \log \frac{n_g(x_i) \wedge n_{g^{\text{target}}}(x_i)}{n_{g^{\text{target}}}(x_i)}$$

$$\leq \sum_{i=1}^{m} \log \frac{n_g(x_i) \wedge 4n_{g^\dagger}(x_i)}{n_{g^\dagger}(x_i) \wedge 4n_g(x_i)} + 2|\neg C| + 2|\neg B| + \sum_{i \in C \cap B} \log \frac{n_{g^\dagger}(x_i)}{n_{g^{\text{target}}}(x_i)} + |\neg C|$$

$$- \sum_{i \in B} \log \frac{n_g(x_i) \wedge n_{g^{\text{target}}}(x_i)}{n_{g^{\text{target}}}(x_i)}$$

$$\leq 18m\bar{r} + 12m\bar{p} + 4. \qquad \text{(Applying Lemmas 4, 5 and 6)}$$

Note that in the second last inequality, we adopt the fact that $\frac{x \wedge 4y}{y \wedge 4x} \in [\frac{1}{4}, 4]$ for all $x, y > 0$. $\qquad \square$

**Lemma 4.** *For any hypothesis $g$ with $\widehat{\ell}^{recall}(g) \leq \bar{r}$ and any subset $S \subset [m]$, we have*

$$\sum_{i \in S \cap B} \log \frac{n_g(x_i) \wedge n_{g^{target}}(x_i)}{n_{g^{target}}(x_i)} \geq -2m\bar{r} - 1.$$

*where $B = \{i | \frac{n_g(x_i)}{n_{g^{target}}(x_i)} \geq \frac{1}{2}\}$.*

*Proof of Lemma 4.* Now let's focus on the rounds in $S \cap B$. We have

$$\sum_{i \in S \cap B} \left(1 - \frac{n_g(x_i) \wedge n_{g^{\text{target}}}(x_i)}{n_{g^{\text{target}}}(x_i)}\right) \leq \sum_{i \in S \cap B} \ell^{\text{recall}}(g, x_i) \leq m\bar{r}.$$

Our problem becomes computing

$$\min \sum_{i \in S \cap B} \log \frac{n_g(x_i) \wedge n_{g^{\text{target}}}(x_i)}{n_{g^{\text{target}}}(x_i)}$$

$$s.t. \sum_{i \in S \cap B} \frac{n_g(x_i) \wedge n_{g^{\text{target}}}(x_i)}{n_{g^{\text{target}}}(x_i)} \geq |S \cap B| - m\bar{r}.$$

By applying Lemma 7, we know

$$\min \sum_{i \in S \cap B} \log \frac{n_g(x_i) \wedge n_{g^{\text{target}}}(x_i)}{n_{g^{\text{target}}}(x_i)} \geq -2m\bar{r} - 1.$$

Thus, we have

$$\sum_{i \in S} \log \frac{n_g(x_i) \wedge n_{g^{\text{target}}}(x_i)}{n_{g^{\text{target}}}(x_i)} \geq \min \sum_{i \in S \cap B} \log \frac{n_g(x_i) \wedge n_{g^{\text{target}}}(x_i)}{n_{g^{\text{target}}}(x_i)} - |\neg B| \geq -4m\bar{r} - 1.$$

$$\square$$

**Lemma 5.** *For any hypothesis $g$ with $\widehat{\ell}^{precision}(g) \leq \overline{p}$ and any subset $S \subset [m]$, we have*

$$\sum_{i \in S \cap A} \log \frac{n_g(x_i)}{n_{g^{target}}(x_i)} \leq 2m\overline{p} + 1 \,.$$

*where $A = \{i | \frac{n_g(x_i)}{n_{g^{target}}(x_i)} \leq 2\}$.*

*Proof of Lemma 5.* Since the empirical precision loss is bounded by $\overline{p}$, we have

$$\sum_{i \in S \cap A} \left( 1 - \frac{n_{g^{\text{target}}}(x_i) \wedge n_g(x_i)}{n_g(x_i)} \right) \leq \sum_{i \in S \cap A} \ell^{\text{precision}}(g, x_i) \leq m\overline{p} \,.$$

By re-arranging terms, we have

$$\sum_{i \in S \cap A} \frac{n_{g^{\text{target}}}(x_i) \wedge n_g(x_i)}{n_g(x_i)} \geq |S \cap A| - m\overline{p} \,.$$

By applying Lemma 7, we have

$$\min \sum_{i \in S \cap A} \log \frac{n_{g^{\text{target}}}(x_i) \wedge n_g(x_i)}{n_g(x_i)} \geq -2m\overline{p} - 1 \,.$$

Hence, we have

$$\sum_{i \in S \cap A} \log \frac{n_g(x_i)}{n_{g^{\text{target}}}(x_i)} \leq \sum_{i \in S \cap A} \log \frac{n_g(x_i)}{n_{g^{\text{target}}}(x_i) \wedge n_g(x_i)} \leq 2m\overline{p} + 1 \,.$$

$\square$

**Lemma 6.** *For any $g$ with $\widehat{\ell}^{recall}(g) \leq \overline{r}$, we have*

$$\sum_i \mathbb{1}\left( \frac{n_g(x_i)}{n_{g^{target}}(x_i)} < \frac{1}{2} \right) < 2m\overline{r} \,.$$

*For any $g$ with $\widehat{\ell}^{precision}(g) \leq \overline{p}$, we have*

$$\sum_i \mathbb{1}\left( \frac{n_g(x_i)}{n_{g^{target}}(x_i)} > 2 \right) < 2m\overline{p} \,.$$

*Proof of Lemma 6.* When $\frac{n_g(x_i)}{n_{g^{\text{target}}}(x_i)} < \frac{1}{2}$, we have

$$\ell^{\text{recall}}(g, x_i) \geq 1 - \frac{n_g(x_i) \wedge n_{g^{\text{target}}}(x_i)}{n_{g^{\text{target}}}(x_i)} > \frac{1}{2} \,.$$

Thus, we have $\sum_i \mathbb{1}\left( \frac{n_g(x_i)}{n_{g^{\text{target}}}(x_i)} < \frac{1}{2} \right) < 2m\overline{r}$. Similarly, when $\frac{n_g(x_i)}{n_{g^{\text{target}}}(x_i)} > 2$, we have

$$\ell^{\text{precision}}(g, x_i) \geq 1 - \frac{n_g(x_i) \wedge n_{g^{\text{target}}}(x_i)}{n_g(x_i)} > \frac{1}{2} \,.$$

Thus, we have $\sum_i \mathbb{1}\left( \frac{n_g(x_i)}{n_{g^{\text{target}}}(x_i)} > 2 \right) < 2m\overline{p}$. $\square$

**Lemma 7.** *For any $k \in \mathbb{N}_+$, $c \geq 0$, let OPT denote the optimal value to the following constrained optimization problem:*

$$\min_{a_{1:k}} \sum_{i=1}^{k} \log a_i$$

$$s.t. \sum_{i=1}^{k} a_i \geq k - c,$$

$$\frac{1}{2} \leq a_i \leq 1, \forall i \in [k] \,.$$

*We have $OPT \geq -2c - 1$.*

*Proof of Lemma 7.* We prove the lemma by showing that in the optimal solution, there will be at most one entry of $a_{1:k}$ not in $\{\frac{1}{2}, 1\}$. In this case, there are $\lfloor 2c \rfloor$ many $\frac{1}{2}$'s and one $c - \frac{\lfloor 2c \rfloor}{2}$. Then, we have

$$\sum_{i=1}^{k} \log a_i \geq -2c - 1.$$

Hence, it suffices to prove that in the optimal solution, there will be at most one entry of $a_{1:k}$ not in $\{\frac{1}{2}, 1\}$. Suppose that there are two entries $a_1 < a_2 \in (\frac{1}{2}, 1)$. For any $\Delta > 0$ s.t. $a_1 - \Delta, a_2 + \Delta \in [\frac{1}{2}, 1]$, we have

$$\log(\frac{a_2 + \Delta}{a_2}) < \log(\frac{a_1}{a_1 - \Delta}),$$

which is due to $\frac{x+\Delta}{x}$ is monotonically decreasing in $x$. By re-arranging terms, we have

$$\log(a_1 - \Delta) + \log(a_2 + \Delta) < \log(a_1) + \log(a_2).$$

Hence, we can always decrease the function value by changing $a_1, a_2$ to $a_1 - \Delta, a_2 + \Delta$. By setting $\Delta = (1 - a_2) \wedge (a_1 - \frac{1}{2})$, either $a_1$ is changed to $\frac{1}{2}$ or $a_2$ is changed to 1. We reduce the number of entries not being $\frac{1}{2}$ or 1. We are done with the proof. $\qquad \square$

### B.3 Surrogate Loss Method in Both Realizable and Agnostic Cases

Again we focus empirical precision and recall losses minimization. Let $x_1, \ldots, x_m \in \mathcal{X}$ denote a sequence of inputs. For each hypothesis $g$, let $U_i^g$ denote the uniform distribution over the output $g(x_i)$. For any pair of hypotheses $g', g''$, define the following:

$$\texttt{precision.loss}(g' \mid g'') = \texttt{recall.loss}(g'' \mid g') := \frac{1}{m} \sum_{i=1}^{m} U_i^{g'}(g'(x_i) \setminus g''(x_i)).$$

Here $\texttt{precision.loss}(g' \mid g'')$ is the precision loss of hypothesis $g'$ when the target hypothesis is $g''$ and $\texttt{recall.loss}(g'' \mid g')$ is the recall loss of hypothesis $g''$ when the target hypothesis is $g'$. Thus, the goal is to output a hypothesis $g$ with small $\texttt{precision.loss}(g \mid g^{\text{target}})$ and $\texttt{recall.loss}(g \mid g^{\text{target}})$.

Our learning rule is based on two simple principles for discarding sub-optimal hypotheses. We illustrate these principles with the following intuitive example: consider a music recommendation system, and assume we are considering two candidate hypotheses, $g_1$ and $g_2$. Both hypotheses recommend classical music; however, $g_1$ recommends pieces by Bach 20% of the time and pieces by Mozart 10% of the time, while $g_2$ never recommends any pieces by Mozart.

Now, suppose that in the training set, users frequently choose to listen to pieces by Mozart. This observation suggests that $g_2$ should be discarded, as it never recommends Mozart. This leads to our first rule: if a hypothesis exhibits a high recall loss, it can be discarded. The second rule addresses precision loss, which is more challenging because it cannot be directly estimated from the data. To illustrate the second rule, imagine that in the training set, users tend to pick Bach pieces only 5% of the time. This suggests that $g_1$ is over-recommending Bach pieces, and therefore, $g_1$ might also be discarded based on its likely precision loss.

We formally capture this using the following metric.

**Definition 1.** *For a hypothesis $g$ define a vector $v_g : \mathcal{H} \times \mathcal{H} \to [0, 1]$ by*

$$v_g(g', g'') = \frac{1}{m} \sum_{i=1}^{m} U_i^g(g'(x_i) \setminus g''(x_i)).$$

*Define a metric $d_{\mathcal{H}}$ between hypotheses by $d_{\mathcal{H}}(g_1, g_2) = \|v_{g_1} - v_{g_2}\|_\infty$.*

Let $\widehat{g}$ be the observed (empirical) hypothesis; i.e. the hypothesis which outputs $\{v_i\}$ at $x_i$ for all $(x_i, v_i)$ in the training set and outputs the empty set for all unobserved input. A standard union bound argument yields:

**Lemma 8.** *Let $g^{target}$ denote the true hypothesis (i.e. the data is generated from $g^{target}$). Then, with probability at least $1 - \delta$:*

$$d_{\mathcal{H}}(\widehat{g}, g^{target}) \leq O\Big(\sqrt{\frac{\log|\mathcal{H}| + \log(1/\delta)}{m}}\Big).$$

We now present our algorithm. We present two variants, one in the realizable setting (when $g^{\text{target}} \in \mathcal{H}$) and one in the general (agnostic) setting.

---

**Algorithm (realizable case):** Let $\varepsilon$ denote the desired error. Output a hypothesis $g^{\text{output}} \in \mathcal{H}$ such that

    1. For all $g \in \mathcal{H}$, $v_{\widehat{g}}(g, g^{\text{output}}) = 0$.

    2. For all $g \in \mathcal{H}$, $v_{g^{\text{output}}}(g^{\text{output}}, g) \geq \varepsilon \implies v_{\widehat{g}}(g^{\text{output}}, g) > 0,$

---

Notice that Item 1 corresponds to the first principle for discarding suboptimal hypotheses described earlier in this section, while Item 2 corresponds to the second principle.

---

**Algorithm (agnostic case):** output a hypothesis $g^{\text{output}} \in \mathcal{H}$ such that

$$d_{\mathcal{H}}(\widehat{g}, g^{\text{output}}) = \min_{g \in \mathcal{H}} d_{\mathcal{H}}(\widehat{g}, g).$$

---

We prove that

**Theorem 8.** *Let $g^{target}$ denote the target hypothesis. Then, for*

$$m = O\Big(\frac{\log|\mathcal{H}| + \log(1/\delta)}{\varepsilon^2}\Big),$$

*the agnostic-case algorithm outputs a hypothesis $g^{output}$ such that with probability at least $1 - \delta$,*

$$\ell^{scalar}(g^{output}) \leq 5 \min_{g \in \mathcal{H}} \ell^{scalar}(g) + \varepsilon.$$

**Remark 2.** *In the realizable setting, our algorithm achieves a quadratic improvement in sample complexity: learning with recall and precision losses at most $\varepsilon$ can be achieved with $O\left(\frac{\log|\mathcal{H}|+\log(1/\delta)}{\varepsilon}\right)$ examples.*

### B.3.1 Proof of Theorem 2

*Proof of Theorem 2.* For simplicity, we adopt payoffs instead of losses here. The payoff of hypothesis $g$ at $x$ is

$$u(g, x) = \frac{|g^{\text{target}}(x) \cap g(x)|}{2|g(x)|} + \frac{|g^{\text{target}}(x) \cap g(x)|}{2|g^{\text{target}}(x)|}.$$

If $g(x) = \emptyset$ and $g^{\text{target}}(x) \neq \emptyset$, $u(g, x) = \frac{1}{2}$; if both are empty set $u(g, x) = 1$. The expected payoff is $u(g) = \mathbb{E}_{x \sim \mathcal{D}}[u(g, x)] = 1 - \ell^{\text{scalar}}(g)$.

**Construction of $g^{\text{target}}$ and $\mathcal{D}$** Let's start by focusing on one single input $x$. There are $n$ items $N_1(x) = [\frac{n}{2}]$ and $N_2(x) = \{\frac{n}{2} + 1, \ldots, n\}$. Consider two hypotheses—$g_1$ with $g_1(x) = N_1(x)$ and $g_2$ with $g_2(x) = N_1(x) \cup N_2(x)$. So $g_1(x)$ contains half of the items in $g_2(x)$.

In a world characterized by $\beta \in [\frac{1}{8}, \frac{2}{3}]$, $g^{\text{target}}(x)$ is generated in the following random way: Randomly select $\frac{3}{4} \cdot \beta n$ items from $N_1(x)$ and $\frac{1}{4} \cdot \beta n$ items from $N_2(x)$. We denote this distribution by $P_\beta$. No matter what $\beta$ is, w.p. $\frac{3}{4}$, $v$ is sampled uniformly at random from $N_1(x)$ and w.p. $\frac{1}{4}$, $v$ is sampled uniformly at random from $N_2(x)$. That is, every item in $N_1(x)$ has probability $\frac{3}{2n}$ of being sampled and every item in $N_2(x)$ has probability $\frac{1}{2n}$ of being sampled.

For any $g^{\text{target}}$ generated from the above process, the payoff of $g_1$ at $x$ is

$$u(g_1, x) = \frac{|g^{\text{target}}(x) \cap g_1(x)|}{2|g_1(x)|} + \frac{|g^{\text{target}}(x) \cap g_1(x)|}{2|g^{\text{target}}(x)|} = \frac{3/4 \cdot \beta n}{n} + \frac{3/4 \cdot \beta n}{2\beta n} = \frac{3}{4}\beta + \frac{3}{8},$$

and the payoff of $g_2$ at $x$ is

$$u(g_2, x) = \frac{|g^{\text{target}}(x) \cap g_2(x)|}{2|g_2(x)|} + \frac{|g^{\text{target}}(x) \cap g_2(x)|}{2|g^{\text{target}}(x)|} = \frac{\beta n}{2n} + \frac{\beta n}{2\beta n} = \frac{1}{2}\beta + \frac{1}{2}.$$

We make infinite copies of $\{x, N_1(x), N_2(x)\}$. In each of the copy, $g_1(x) = N_1(x)$ and $g_2(x) = N_1(x) \cup N_2(x)$. For each $x$, we independently sample $g^{\text{target}}(x)$ from $P_\beta$. Let the data distribution over all of such copies of $x$. Then almost surely, there is no repentance in the training data, i.e., there does not exist $i \neq j$ such that $x_i = x_j$. And for any random sampled test point, w.p. 1, it has not been sampled in the training set.

**Analysis** For any unobserved $x \notin \{x_i | i \in [m]\}$, let $\alpha_1 = \frac{|g^{\text{output}}(x) \cap N_1(x)|}{n}$ and $\alpha_2 = \frac{|g^{\text{output}}(x) \cap N_2(x)|}{n}$. Note that $\alpha_1, \alpha_2$ are in $[0, \frac{1}{2}]$ and are possibly random variables if $\mathcal{A}$ is randomized. Then the expected (over the randomness of $g^{\text{target}}$) payoff of $g^{\text{output}}$ at $x$ is

$$\begin{aligned}
\mathbb{E}_{g^{\text{target}}}\left[u(g^{\text{output}}, x)\right] &= \mathbb{E}_{g^{\text{target}}}\left[\frac{|g^{\text{target}}(x) \cap g^{\text{output}}(x)|}{2|g^{\text{output}}(x)|} + \frac{|g^{\text{target}}(x) \cap g^{\text{output}}(x)|}{2|g^{\text{target}}(x)|}\right] \\
&= \frac{\alpha_1 n \cdot \frac{3}{2}\beta + \alpha_2 n \cdot \frac{1}{2}\beta}{2(\alpha_1 + \alpha_2)n} + \frac{\alpha_1 n \cdot \frac{3}{2}\beta + \alpha_2 n \cdot \frac{1}{2}\beta}{2\beta n} \\
&= \frac{\alpha_1 \beta}{2(\alpha_1 + \alpha_2)} + \frac{\beta}{4} + \frac{3}{4}\alpha_1 + \frac{1}{4}\alpha_2,
\end{aligned} \tag{5}$$

which is monotonically increasing in $\alpha_1$. Hence $\mathbb{E}_{g^{\text{target}}}\left[u(g^{\text{output}}, x)\right]$ is maximized at $\alpha_1 = \frac{1}{2}$. Then

$$\mathbb{E}_{g^{\text{target}}}\left[u(g^{\text{output}}, x)\right] \leq \frac{\beta}{4} \cdot \left(\frac{1}{\frac{1}{2} + \alpha_2} + 1\right) + \frac{3}{8} + \frac{1}{4}\alpha_2.$$

Note that $\beta$ is not observable if we never sample the same $x$ more than once (and thus the distribution of $v$ conditional on $\beta$ is identical for any $\beta$). Hence $g^{\text{output}}$ is independent of $\beta$.

- If $\mathcal{P}_{x \sim \mathcal{D}}(\alpha_2(x) \leq \frac{1}{4}) \geq \frac{1}{2}$: when $\beta = \frac{1}{8}$, $\mathbb{E}_{g^{\text{target}}}\left[u(g^{\text{output}}, x)\right] \leq \frac{1}{4}(\frac{1}{4+8\alpha_2} + \alpha_2) + \frac{13}{32}$ is monotonically increasing in $\alpha_2$. Hence,

$$u(g_2) - \mathbb{E}_{g^{\text{target}}}\left[u(g^{\text{output}})\right] \geq \frac{1}{2}\left(\frac{9}{16} - \frac{49}{96}\right) = \frac{5}{192} = \frac{5}{84}\ell^{\text{scalar}}(g_2).$$

- If $\mathcal{P}_{x \sim \mathcal{D}}(\alpha_2(x) > \frac{1}{4}) \geq \frac{1}{2}$: when $\beta = \frac{2}{3}$, $\mathbb{E}_{g^{\text{target}}}\left[u(g^{\text{output}}, x)\right] = \frac{1}{3+6\alpha_2} + \frac{1}{4}\alpha_2 + \frac{13}{24}$ is maximized at $\alpha_2 = \frac{1}{2}$ for $\alpha \in [\frac{1}{4}, \frac{1}{2}]$. Hence,

$$u(g_1) - \mathbb{E}_{g^{\text{target}}}\left[u(g^{\text{output}})\right] \geq \frac{1}{2}\left(\frac{7}{8} - \frac{5}{6}\right) = \frac{1}{48} = \frac{1}{6}\ell^{\text{scalar}}(g_1).$$

Therefore, for any algorithm $\mathcal{A}$, for any $x_{1:m}, v_{1:m}$, $\mathbb{E}_{g^{\text{target}}}\left[\ell^{\text{scalar}}(g^{\text{output}})\right]$ is worse than $1.05 \cdot \min\{\ell^{\text{scalar}}(g_1), \ell^{\text{scalar}}(g_2)\}$ at either $\beta = \frac{1}{8}$ or $\beta = \frac{2}{3}$. So there exists a target hypothesis such that $\ell^{\text{scalar}}(g^{\text{output}}) \geq 1.05 \cdot \min\{\ell^{\text{scalar}}(g_1), \ell^{\text{scalar}}(g_2)\}$. $\square$

### B.3.2 Proof of Theorem 8

We use the following auxiliary metric between hypotheses:

**Definition 2.** *For two hypotheses $g', g''$ define*

$$\begin{aligned}
d_{\text{p,r}}(g', g'') &= \texttt{precision.loss}(g'|g'') + \texttt{recall.loss}(g'|g'') \\
&= \texttt{precision.loss}(g''|g') + \texttt{recall.loss}(g''|g').
\end{aligned}$$

For any hypothesis $g$, the scaler loss $\ell^{\text{scalar}}(g) = \frac{1}{2}d_{\text{p,r}}(g^{\text{output}}, g^{\text{target}})$. In the remainder of this section, we focus on proving Theorem 8. The basic idea is to show that $d_{\mathcal{H}}$ can be used as a surrogate for $d_{\text{p,r}}$. The following lemma plays a crucial role in our proof.

**Lemma 9.** *For every pair of hypotheses $g_1, g_2$:*

$$d_{\mathcal{H}}(g_1, g_2) \le d_{\text{p,r}}(g_1, g_2).$$

*If in addition $g_1, g_2 \in \mathcal{H}$, we have:*

$$d_{\text{p,r}}(g_1, g_2) \le 2d_{\mathcal{H}}(g_1, g_2).$$

We first use Lemma 9 to prove Theorem 8, and later prove the Lemma.

*Proof of Theorem 8.* Assume $m = O(\frac{\log|\mathcal{H}| + \log(1/\delta)}{\varepsilon^2})$ is such that $d_{\mathcal{H}}(g^{\text{output}}, g^{\text{target}}) \le \varepsilon/4$ with probability at least $1 - \delta$, and assume the latter event holds. Let $g \in \mathcal{H}$, by the triangle inequality:

$$d_{\text{p,r}}(g^{\text{output}}, g^{\text{target}}) \le d_{\text{p,r}}(g^{\text{output}}, g) + d_{\text{p,r}}(g, g^{\text{target}}).$$

We upper bound the first term on the right-hand side as follows:

$$
\begin{aligned}
d_{\text{p,r}}(g^{\text{output}}, g) &\le 2d_{\mathcal{H}}(g^{\text{output}}, g) && \text{(Lemma 9)} \\
&\le 2d_{\mathcal{H}}(g^{\text{output}}, g^{\text{target}}) + 2d_{\mathcal{H}}(g^{\text{target}}, g) && \\
&\le 4d_{\mathcal{H}}(g^{\text{target}}, g) + \varepsilon && \text{(see below)} \\
&\le 4d_{\text{p,r}}(g^{\text{target}}, g) + \varepsilon. && \text{(Lemma 9)}
\end{aligned}
$$

Altogether,

$$d_{\text{p,r}}(g^{\text{output}}, g^{\text{target}}) \le 5d_{\text{p,r}}(g, g^{\text{target}}) + \varepsilon.$$

It remains to explain the second to last inequality above. It follows by two applications of the triangle inequality:

$$
\begin{aligned}
d_{\mathcal{H}}(g^{\text{output}}, g^{\text{target}}) &\le d_{\mathcal{H}}(g^{\text{output}}, \widehat{g}) + \varepsilon/4 && (d_{\mathcal{H}}(g^{\text{target}}, \widehat{g}) \le \varepsilon/4) \\
&\le d_{\mathcal{H}}(g, \widehat{g}) + \varepsilon/4 && (g^{\text{output}} \in \arg\min_{g \in \mathcal{H}} d_{\mathcal{H}}(g, \widehat{g})) \\
&\le d_{\mathcal{H}}(g, g^{\text{target}}) + \varepsilon/2. && (d_{\mathcal{H}}(g^{\text{target}}, \widehat{g}) \le \varepsilon/4)
\end{aligned}
$$

$\square$

*Proof of Lemma 9.* For the first inequality, note that both of the distributions $U_i^{g_1}$ and $U_i^{g_2}$ are uniform over their supports and hence $\text{TV}(U_i^{g_1}, U_i^{g_2}) = \max\{U_i^{g_1}(g_1(x_i) \setminus g_2(x_i)), U_i^{g_2}(g_2(x_i) \setminus g_1(x_i))\}$. Thus, for every $g', g'' \in \mathcal{H}$:

$$
\begin{aligned}
&|U_i^{g_1}(g'(x_i) \setminus g''(x_i)) - U_i^{g_2}(g'(x_i) \setminus g''(x_i))| \\
\le & \text{TV}(U_i^{g_1}, U_i^{g_2}) \\
\le & U_i^{g_1}(g_1(x_i) \setminus g_2(x_i)) + U_i^{g_2}(g_2(x_i) \setminus g_1(x_i)).
\end{aligned}
$$

Hence, by averaging the above inequalities over $i = 1, \ldots, n$:

$$d_{\mathcal{H}}(g_1, g_2) \le \frac{1}{m}\sum_{i=1}^{m} \text{TV}(U_i^{g_1}, U_i^{g_2}) \le d_{\text{p,r}}(g_1, g_2),$$

which yields the first inequality.

For the second inequality, assume $g_1, g_2 \in \mathcal{H}$. Thus,

$$
\begin{aligned}
d_{\mathcal{H}}(g_1, g_2) &\ge \max\left\{\frac{1}{m}\sum_{i=1}^{m} U_i^{g_1}(g_1(x_i) \setminus g_2(x_i)), \frac{1}{m}\sum_{i=1}^{m} U_i^{g_2}(g_2(x_i) \setminus g_1(x_i))\right\} \\
&\ge \frac{1}{m}\sum_{i=1}^{m} \frac{U_i^{g_1}(g_1(x_i) \setminus g_2(x_i)) + U_i^{g_2}(g_2(x_i) \setminus g_1(x_i))}{2} \\
&= \frac{1}{2}d_{\text{p,r}}(g_1, g_2).
\end{aligned}
$$

$\square$

## B.4 Algorithm and Proofs in the Semi-Realizable Case

In the semi-realizable case, there exists a hypothesis in the class with zero precision loss. The question is whether we achieve zero precision loss while allowing for the worst recall loss in the class.

**Theorem 5.** *There exists an algorithm such that if there exists a hypothesis $g' \in \mathcal{H}$ with $\ell^{precision}(g') = 0$ and $\ell^{recall}(g') = r$, then given an IID training set of size $O(\frac{\log(|\mathcal{H}|/\delta)}{\Delta_{\mathcal{D}}^2})$, with probability $1 - \delta$, it outputs a hypothesis with $\ell^{precision}(g^{output}) = 0$ and $\ell^{recall}(g^{output}) = r$.*

The algorithm works as follows.

---

**Algorithm:** output

$$g^{\text{output}} = \arg\min_{g \in \mathcal{H}} \sum_{i=1}^{m} \frac{\mathbb{1}(v_i \in g(x_i))}{n_g(x_i)} .$$

If there are multiple solutions, we break ties by picking the hypothesis with smallest empirical recall loss.

---

*Proof of Theorem 5.* For any hypothesis $g$, $\mathbb{1}(v_i \in g(x_i))$ is an unbiased estimate of the recall $\frac{|g(x_i) \cap g^{\text{target}}(x_i)|}{n_{g^{\text{target}}}(x_i)}$. Thus, $\frac{\mathbb{1}(v_i \in g(x_i))}{n_g(x_i)}$ is an unbiased estimate of $\frac{|g(x_i) \cap g^{\text{target}}(x_i)|}{n_g(x_i) \cdot n_{g^{\text{target}}}(x_i)}$.

Since $g'$ has zero precision loss, $\frac{|g(x_i) \cap g^{\text{target}}(x_i)|}{n_g(x_i)} = 1$ almost everywhere. Thus, we have

$$\left| \frac{1}{m} \sum_{i=1}^{m} \frac{\mathbb{1}(v_i \in g(x_i))}{n_g(x_i)} - \mathbb{E}\left[ \frac{1}{n_{g^{\text{target}}}(x)} \right] \right| \leq \sqrt{\frac{\log(|\mathcal{H}|) + \log(1/\delta)}{m}} ,$$

for all $g \in \mathcal{H}$. Then if $\Delta_{\mathcal{D}} > 0$, we need $\frac{1}{\Delta_{\mathcal{D}}^2}$ samples to separate $g'$ from other hypotheses in the hypothesis class. $\square$

**Theorem 6.** *There exists a class $\mathcal{H} = \{g_1, g_2\}$ of two hypotheses, for any $m > 0$ and any (possibly randomized improper) algorithm $\mathcal{A}$, there exists a target hypothesis $g^{target}$ and a data distribution $\mathcal{D}$ for which there exists a hypothesis $g^{\dagger} \in \mathcal{H}$ with $\ell^{precision}(g^{\dagger}) = 0$ s.t. with probability $1 - \delta$ over the training set, the expected (over the randomness of the algorithm) precision and recall losses of the output $g^{output}$ satisfy either $\mathbb{E}\left[\ell^{recall}(g^{output})\right] \geq \min_{g \in \mathcal{H}} \ell^{recall}(g) + \Omega(1)$ or $\mathbb{E}\left[\ell^{precision}(g^{output})\right] = \Omega(1)$.*

*Proof of Theorem 6.* Let's start by focusing on one single input $x$. Let $N = \{v_1, \ldots, v_n\}$ for some $n \gg m$. Let $g_1(x) = \{v_1\}$ and $g_2(x) = \{v_2\}$. In world I, $g^{\text{target}}(x)$ is generated in the following way.

- w.p. $\frac{1}{2}$, $g^{\text{target}}(x) = N \setminus \{v_2\}$.

- w.p. $\frac{1}{2}$, $g^{\text{target}}(x) = \{v_1, v_2\}$.

We construct a symmetric world II by switching $v_1$ and $v_2$, i.e.,

- w.p. $\frac{1}{2}$, $g^{\text{target}}(x) = N \setminus \{v_1\}$.

- w.p. $\frac{1}{2}$, $g^{\text{target}}(x) = \{v_1, v_2\}$.

We make infinite independent copies of $(x, N)$ and let $\mathcal{D}$ to be the uniform distribution over such $x$'s. Hence, in world I, $\ell^{\text{precision}}(g_1) = 0$ and $\ell^{\text{recall}}(g_1) = \frac{3}{4} - \frac{1}{2n}$; $\ell^{\text{precision}}(g_2) = \frac{1}{2}$ and $\ell^{\text{recall}}(g_2) = \frac{3}{4}$. When $n \to \infty$, we can't distinguish between two worlds. In order to achieve $\ell^{\text{precision}}(g^{\text{output}}) = 0$ in both worlds, we need to make $g^{\text{output}}(x) = \emptyset$ for almost every $x$. Then the recall loss would be 1. $\square$

