# OpenReview forum: "Probably Approximately Precision and Recall Learning"
_NeurIPS.cc/2025/Conference — NeurIPS 2025 poster_

### Official Review · Reviewer_sqyN · 2025-06-22

**Clarity:** 3
**Significance:** 4
**Originality:** 4
**Rating:** 4
**Confidence:** 3

**Summary:**

As traditional learning paradigms fail when only positive labels are observed, a new PAC-learning framework is presented to optimize precision and recall losses under partial feedback. This setup in practice applies to multi-label learning tasks, such as tagging people in photos, where only some tagged names are observed, but not the full set of people.  The paper shows that optimizing precision is fundamentally more difficult than optimizing recall in this setting because of the absence of negative feedback. Two algorithms are presented, the maximum likelihood-based algorithm, and the surrogate loss minimization algorithm, which are shown to achieve upper and lower bounds. Two settings are analyzed: i. Realizable Setting, when the true hypothesis is in class H; ii. Agnostic Setting, when the true hypothesis not necessarily in H.

**Questions:**

1.	On Practical Impact and Real-World Feasibility: Could the authors elaborate more on how their algorithms might be adapted to real-world tasks (e.g., recommender systems or tagging applications)? If possible, implementation on some toy datasets, including some commentary on feasibility under partial observability assumptions. These would strengthen the paper’s significance.
2.	Clarification on Output Size Assumptions: Several results depend on the assumption that the output size n_G^{target} is bounded or known. In many practical applications, this is not directly available. Could the authors discuss how their methods behave when this size is unknown or only loosely bounded?
3.	Intuition and Use Cases for Scalar vs. Pareto Loss: The distinction between scalar-loss and Pareto-loss objectives is theoretically clear. Could the authors clarify what kind of applications might naturally benefit more from optimizing for one versus the other?
4.	Lower Bounds Interpretation: The paper proves lower bounds like \alpha ≥1.05 for the scalar loss and impossibility of achieving additive error for Pareto loss. Could the authors comment on whether these lower bounds are tight in any known cases? Are there scenarios where existing heuristics approach these bounds?
5.	Improving Accessibility: While the main ideas are well-explained, could the authors consider adding more intuitive commentary or diagrams near the formal results to guide non-theory readers (e.g., explaining the significance of each key theorem in words, or linking them to the motivating examples)?

**Ethical Concerns:**

["NO or VERY MINOR ethics concerns only"]

**Limitations:**

Authors only discussed the theoretical limitations of the paper. As this is almost purely a theoretical paper, the authors did not discuss any societal impact aspects of this work.

**Paper Formatting Concerns:**

Did not notice any issues.

**Quality:**

3

**Strengths And Weaknesses:**

Strengths:
•	Originality: The paper tackles a novel and important learning setting, which is learning under one-sided (positive-only) feedback. This is highly relevant in real-world applications like multi-label learning and recommender systems. The formalization of this setup into a new PAC-learning framework is a meaningful and original contribution.
•	Significance: By highlighting fundamental differences between learning precision and recall under partial supervision, the work sheds light on overlooked aspects of supervised learning theory. In particular, the hardness of estimating precision loss without negative labels is both practically and theoretically significant.
•	Theoretical Quality: The paper demonstrates a high level of rigor in constructing new learning guarantees. It provides both upper and lower bounds, separating realizable and agnostic settings, and analyzes both scalar-loss and Pareto-loss objectives. These contributions form a solid theoretical foundation.
•	Algorithmic Contributions: Two learning algorithms, based on maximum likelihood estimation and surrogate loss minimization, are clearly described and motivated. They show that learning from only positive feedback is tractable in some settings, but fundamentally limited in others.
•	Clarity of High-Level Ideas: Despite the complex theoretical nature, the paper does a good job of motivating the problem and building intuition through illustrative examples (e.g., the indistinguishable hypotheses with vastly different precision losses).
Weaknesses:
•	Clarity for Non-Theorists: While the high-level concepts are well-motivated, the technical sections, especially the proof sketches may be hard to parse for readers not familiar with PAC learning theory or sample complexity analysis. A bit more intuitive commentary around key lemmas and theorems would help make the results more broadly accessible.
•	No Empirical Results: The paper is entirely theoretical. Although this is acceptable for NeurIPS theory tracks, some small-scale synthetic experiments (even toy data) could help reinforce the practical relevance and intuition behind the proposed hardness results or algorithmic strategies.
•	Assumptions on Output Size Knowledge: Some results depend on knowing or bounding the output size n_G^{target}, which may be non-trivial in practice. The practical implications of this assumption are not fully explored.
•	Practical Impact Path Not Fully Developed: Although the authors claim relevance to real-world domains, there is a gap between the theory and potential practical implementation guidance. It would be helpful to bridge that more directly, even if via discussion or future work.

---

> ### Author Rebuttal · Authors · 2025-07-30
>
> We thank the reviewer for their positive and insightful feedback. In what follows, we address comments and questions.
>
> - **Clarity for Non-Theorists:** Thank you for the feedback. We will add more details and provide intuition to clarify the proofs for future readers.
>
> - **Assumptions on Output Size Knowledge**: We only assume that the output size of the target hypothesis is **bounded**, but not necessarily known, in Theorem 5. This mild assumption enables us to achieve zero precision loss in the semi-realizable setting. We complement this with the **impossibility result in Theorem 6**, which shows that without any such bound on the output size, it is impossible to achieve an additive guarantee on precision loss. Together, these results characterize the role of output size in a sharp and principled way.
>
> - **On Practical Impact and Real-World Feasibility**:
> Thank you for this suggestion. As noted, our paper is intended as a theoretical contribution—specifically, it defines an analog of PAC learning for precision and recall—focused on rigorously modeling learning problems with asymmetric loss under partial feedback. We introduced a general framework for studying such problems, proved basic results about the statistical expressiveness of classical algorithmic templates (such as ERM and MLE), and highlighted key algorithmic and information-theoretic phenomena that arise in this setting. Our goal is to lay a *theoretical foundation* for precision/recall learning under partial observability. While adapting these ideas into concrete implementations and empirical studies is an important direction, it lies beyond the scope of this work. We hope that the framework we propose will serve as a basis for future research into practical algorithms that operate under similar constraints.
>
> - **Intuition and Use Cases for Scalar vs. Pareto Loss**: The scalar loss is appropriate when there’s a known trade-off between precision and recall—for example, in medical screening, one may weigh recall more heavily to avoid missing true positives.
> The Pareto loss, on the other hand, is useful when one metric must meet a strict operational threshold while optimizing the other. For instance, in spam detection, a system may be required to maintain precision above 95% to avoid frustrating users with false positives. Within that constraint, the goal is to maximize recall. Such asymmetric constraints cannot be naturally encoded by a fixed convex combination, making Pareto-based optimization the more appropriate tool in these scenarios.
>
> - **Lower Bounds Interpretation**: We prove a lower bound showing that no algorithm can achieve a multiplicative factor smaller than 1.05 for scalar loss, and we give an upper bound of 5 using our surrogate loss approach. It remains an open question whether this gap can be closed, and identifying the tight constant is, in our view, an interesting technical challenge. That said, we want to emphasize that the key takeaway from our lower bound is *qualitative rather than quantitative*: it establishes that achieving an *additive* error bound is fundamentally impossible in this setting. This marks a sharp departure from standard PAC learning and highlights the need for new learning principles under partial feedback.
>
> - **Improving Accessibility**: We will add more intuitive explanations after each result.

---

### Official Review · Reviewer_k9sU · 2025-07-02

**Clarity:** 2
**Significance:** 2
**Originality:** 2
**Rating:** 2
**Confidence:** 3

**Summary:**

This work focuses on a learning setting where the learning data consists of partial feedback and the goal is to learn to predict a set of labels. Specifically, it considers two settings, realizable setting and an agnostic setting, depending on whether the ground truth hypothesis is in the hypothesis class. It further presents the theoretical results on the learnability for both settings, including both positive and negative results.

**Questions:**

In addition to those raised in [Strengths And Weaknesses]:
- why obtaining an unbiased estimation of precision loss is not possible in this setting (line 216)? My understanding is that as long as the sample is large enough, all the target labels would be observed.
- at line 100, it is mentioned that in the partial feedback setting, it is impossible to estimate precision loss while later the scalar-loss objective is defined as the combination of recall and precision loss. Can the authors clarify on what metrics can be computed exactly and if there are any approximations?
- what's the computational complexity of algorithm 1?

**Ethical Concerns:**

["NO or VERY MINOR ethics concerns only"]

**Final Justification:**

As mentioned in the discussion, I still don't see how the theoretical insights lead to a practical algorithm, and there's a lack of empirical results. Since the authors didn't reply to my concerns, I'll keep my score.

**Limitations:**

Yes.

**Paper Formatting Concerns:**

None.

**Quality:**

2

**Strengths And Weaknesses:**

The problem of learning from partial labels considered in this work is interesting since such scenarios can happen when labeling is costly. However, a few concerns arise when reading this work:
- the assumption that the hypothesis class is finite weakens the theoretical results in this work since it doesn't hold in practice.
- missing related work: the problem setting is closely related to the PU learning setting [1,2] where only positive labels are observed.
- in this work, it compares with ERM a lot but it's unclear to me how ERM is defined in this work and why ERM would fail in this case. In the paragraph from Line 121, the authors claim that the reason ERM would fail is because two hypotheses can have the same recall loss but different precision loss, which is further illustrated by Example 1. However, in example 1, recall losses are not impossible to compute since it requires the ground truth label set {u_1, ..., u_n}, which is not accessible since the training datapoints only reveal one label, while only precision loss is possible. Further, precision loss can actually distinguish the two hypotheses in this case, so ERM could possibly work. Can the authors elaborate more on this?
- Theorem only proves about PAC learnability but not the efficiency of the proposed algorithm. For the first algorithm maximum likelihood, the idea is standard and not really a technical contribution; also, it doesn't seem that the optimization problem can be efficiently solved.
- the definitions of G' and G'' in algorithm 2 are unclear. Can the authors provide an example to show how this vector is computed?
- minor: at line 189, should the domain be X*Y instead of X^2?

[1] Bekker, Jessa, and Jesse Davis. "Learning from positive and unlabeled data: A survey." Machine Learning 109.4 (2020): 719-760.
[2] Xu, Yixing, et al. "Multi-Positive and Unlabeled Learning." IJCAI. 2017.

---

> ### Author Rebuttal · Authors · 2025-07-30
>
> We thank the reviewer for their feedback. In what follows, we address comments and questions.
>
>
> **Finite hypothesis class**
>
> Note that in many real-world applications, the feature space is effectively finite—for instance, in image classification, where each input is a fixed-size array of pixels, or in recommender systems and user modeling, where individuals are represented by a finite collection of features. In such cases, the log |H| bounds we provide can offer meaningful performance guarantees. Additionally, in many natural parametrized classes, bounds for infinite hypothesis classes can often be derived by reducing to the finite case via a suitable covering argument. For example, in geometric settings, the cover size typically scales as (1/ε)^d, where d is a natural notion of dimension—yielding bounds that scale with d and an additional log(1/ε) factor.
>
> In addition, many of our results—particularly the negative ones—highlight structural barriers that are not tied to the assumption of finiteness. For example, we show that ERM fails while MLE succeeds, and that additive excess risk bounds are unachievable in the agnostic setting already for finite classes.
>
> For the positive results, we agree that it is interesting to extend the analysis to infinite classes, potentially via combinatorial parameters analogous to the VC dimension. This remains an open direction (see also the discussion section and Footnote 2), and we hope our work helps identify the right questions—for instance, whether learnability in this setting can be characterized by a new type of dimension.
>
>
> **PU learning**
>
> Thanks. We’ll add a discussion in the related work on PU learning.
> In particular, while there are only positive examples in the PU model, the unlabeled examples provide significant information about the (potentially) negative examples.
>
> **Why ERM fails**
>
> When we refer to **ERM** in this work, in the realized setting, we mean the selection of an arbitrary hypothesis that is consistent with the observed data. Specifically, since the training data only reveals one label $v_i$ per input $x_i$, we define ERM as selecting
> $$
> \arg\min_{G \in \mathcal{H}} \sum_{i=1}^m \mathbf{1}(v_i \notin G(x_i)).
> $$
>
> As noted in Line 190, the observed label $v_i$ is drawn uniformly at random from the ground-truth set $G_{\text{target}}(x_i)$.
>
> In **Example 1**, $G_2(x_i) = \{u’\}$ does not contain any correct label, so $v_i \notin G_2(x_i)$ for all $i$. $G_1(x_i) = \{u_n\}$ contains onecorrect label $u_n$, but when the size of the ground-truth set $n$ is large, the probability that the observed $v_i$ matches this single predicted label $u_n$ is vanishingly small. With high probability, for all $i$, we will observe a label $v_i$ that does *not* belong to either $G_1(x_i)$ or $G_2(x_i)$, causing both to incur maximal empirical loss (i.e., a loss of 1 for each $i$).
>
> Thus, ERM cannot distinguish between them based on the training data, even though $G_1$ has perfect precision and $G_2$ has zero precision. This illustrates that ERM fails to identify the better hypothesis in this setting.
>
> **Computational complexity**
>
> Our focus in this work is on **statistical complexity**, not computational efficiency. The central questions we aim to answer are: (1) whether learning with precision and recall is *statistically* possible under positive feedback, and (2) how many samples are needed to achieve this. From an algorithmic standpoint, we also investigate whether classical ERM works in this setting—and show that it does not—and identify alternative principles that do, such as maximum likelihood.
>
> While the idea of using maximum likelihood (MLE) is standard, its behavior in this partial-information setting is non-trivial and has not been analyzed before.
>
> Moreover, establishing the applicability of a classical and intuitive principle like MLE is an advantage—especially since the more naive ERM approach fails in our setting. More broadly, we see theoretical value in rigorously analyzing simple and natural algorithms, particularly when the analysis is non-trivial. In our view, this makes the result even more interesting.
>
> We view our contribution as highlighting its effectiveness in a regime where ERM provably fails. That said, the question of *efficient* optimization is indeed important, and we see it as an important direction for future work.
>
> **Definitions of G' and G'' in algorithm 2**
>
> $G'$ and $G''$ are not special hypotheses—they are simply elements of the hypothesis class $\mathcal{H}$ and serve as inputs to the definition of the vector $v_G: \mathcal{H} \times \mathcal{H} \to [0,1]$.
>
> Each entry of the vector $v_G$ corresponds to a pair $(G', G'') \in \mathcal{H} \times \mathcal{H}$ and can be written explicitly as
> $$
> v_G(G', G'') = \frac{1}{m} \sum_{i=1}^m \frac{|G'(x_i) \setminus G''(x_i)|}{n_G(x_i)},
> $$
> where $n_G(x_i) = |G(x_i)|$.
> We will clarify this in the revised version.
>
> **Why obtaining an unbiased estimation of precision loss is not possible**
>
> We’d like to clarify that in our setting, the inputs $x_i$ are i.i.d. samples from a data distribution with potentially infinite support. Each input $x_i$ is associated with a ground-truth label set $G_{\text{target}}(x_i)$, but we observe only a single label $v_i$, drawn uniformly at random from $G_{\text{target}}(x_i)$.
> As a result, each $x_i$ is typically seen only once, and we do not observe the full set $G_{\text{target}}(x_i)$. This makes it fundamentally impossible to determine whether the *unobserved* labels predicted by a hypothesis are correct. Therefore, we cannot obtain an unbiased estimate of precision loss under this partial feedback model.
>
> **``at line 100…''**
>
> Thanks for the question. To clarify: in our setting, **recall loss can be estimated with vanishing additive error**, because we observe whether the sampled label $v_i$ is included in the predicted set $G(x_i)$. However, **precision loss cannot be estimated in an unbiased way**, since we do not know which predicted labels (other than the observed $v_i$) are actually correct or incorrect.
>
> Despite this, we show that in the **realizable setting**, where there exists a hypothesis with both zero precision and recall loss, it is still possible to identify such a hypothesis using the MLE-based algorithm—even without being able to directly estimate precision loss.
>
> In contrast, in the **semi-realizable setting**, where there exists a hypothesis with zero precision loss but non-zero recall loss, we prove that no algorithm can achieve both vanishing precision and optimal recall loss (see Theorem 6). This demonstrates the inherent difficulty of optimizing precision under partial feedback, especially outside the realizable case.
>
> **Computational complexity of algorithm 1**
>
> Like MLE in other learning settings, Algorithm 1 requires evaluating a score for each hypothesis in $\mathcal{H}$ based on the training data. This results in a total computational cost of $O(m |\mathcal{H}|)$, where $m$ is the number of samples.

---

> > ### Comment · Reviewer_k9sU · 2025-08-05
> >
> > Thanks for the clarification. While I appreciate the theoretical contributions, I still don't see how the theoretical insights lead to a practical algorithm, and also, the concern that there's a lack of empirical results remains. I'll keep my score.

---

### Official Review · Reviewer_hDwa · 2025-07-03

**Clarity:** 2
**Significance:** 3
**Originality:** 3
**Rating:** 4
**Confidence:** 3

**Summary:**

This paper introduces a novel PAC learning framework for precision-recall optimization under partial feedback, where algorithms observe only positive examples during training (e.g., a single randomly sampled label from the ground-truth set) but must predict complete label sets at test time. The authors demonstrate that classical approaches like Empirical Risk Minimization provably fail in this setting, even for simple hypothesis classes, due to the inability to estimate precision loss without negative examples. They propose two new algorithms for the realizable case achieving optimal $O(\log|H|/\epsilon)$ sample complexity, and establish fundamental limitations in the agnostic setting where additive approximation guarantees are impossible, requiring multiplicative approximation factors instead.

**Questions:**

See weaknesses.

**Ethical Concerns:**

["NO or VERY MINOR ethics concerns only"]

**Limitations:**

the authors adequately addressed the limitations and potential negative societal impact

**Paper Formatting Concerns:**

no major formatting issues

**Quality:**

3

**Strengths And Weaknesses:**

Strengths:
This paper proposes an interesting problem: how can models optimize to achieve good precision and recall when only partial positive labels are available, which is meaningful in real-world applications.

Weaknesses:
The paper has certain relevance to "Revisiting Pseudo-Label for Single-Positive Multi-Label Learning". That paper proves the learnability under single-positive multi-label learning through ERM and proposes two assumptions (the assumption of pseudo-label accuracy and the assumption of data distribution sampling). The authors should provide some discussion regarding that paper. In addition, the related work section should include discussion of articles related to multi-label learning with missing labels, such as [1-5].
[1] Revisiting Pseudo-Label for Single-Positive Multi-Label Learning
[2] Multi-Label Learning from Single Positive Labels
[3] Label-Aware Global Consistency for Multi-Label Learning with Single Positive Labels
[4] One Positive Label is Sufficient: Single-Positive Multi-Label Learning with Label Enhancement
[5] Large Loss Matters in Weakly Supervised Multi-Label Classification

---

> ### Author Rebuttal · Authors · 2025-07-29
>
> We thank the reviewer for their helpful feedback and for bringing these references to our attention. We agree that discussing these papers would strengthen our related work section, and we will gladly add an appropriate discussion of these works in the revised version.

---

> > ### Comment · Reviewer_hDwa · 2025-08-05
> >
> > Thank you for the authors’ response. After reviewing the other reviewers’ comments, I have decided to maintain my original score.

---

### Official Review · Reviewer_PCHe · 2025-07-05

**Clarity:** 3
**Significance:** 3
**Originality:** 3
**Rating:** 5
**Confidence:** 3

**Summary:**

This paper studies PAC learning for the scenario where the hypotheses are set functions that map each input to a set of labels, instead of traditional single-label predictions. The authors consider realizable and agnostic settings. For realizable setting, they prove the sample complexity upper bound $O(\frac{\log|H|/\delta}{\epsilon})$ to achieve both $\epsilon$ precision and recall rate. They give two algorithms to achieve this rate: one is maximum likelihood estimation and the other is minimizing surrogate loss. In the agnostic setting, they prove that achieving additive error is impossible by providing two lower bounds. Instead, they prove the sample complexity to achieve a multiplicative excess risk. The algorithm achieving this rate is the adapted version of  the realizable setting algorithms to the agnostic case.

**Questions:**

Please see the weakness part.

**Ethical Concerns:**

["NO or VERY MINOR ethics concerns only"]

**Final Justification:**

In rebuttal, the authors discussed the potential solution to my question about the constant in upper and lower bound. This is a good theory paper and I recommend acceptance.

**Limitations:**

This is a theory paper so I did not see potential negative societal impact.

**Quality:**

3

**Strengths And Weaknesses:**

Strength:

This paper for the first time studies the PAC learnability of precison and recall losss, which is very novel and useful, since in practice, people care more about PR rate of an algorithm rather than the overall misclassification rate. The theory is neat and interesting, and the algorithms are intuitive and natural---the authors show that the traditional ERM will fail but MLE is a PAC learner for precision and recall loss. They further provide impossibility results for agnostic setting, which match with my intuition that one cannot do well both in precision and recall rate.

Weaknesses:
The authors claim that the additive excess risk bound is not achievable, and a multiplicative factor is unavoidable. However, the constant in lower bound and upper bound is rather large (1.05 vs 5). Hence I am curious what will be the optimal constant. Is that achievable by refining your techniques?

---

> ### Author Rebuttal · Authors · 2025-07-29
>
> We thank the reviewer for their positive and insightful feedback.
>
> Regarding the optimal multiplicative constant: we agree that precisely characterizing this constant is an intriguing open problem, which we leave for future exploration.
>
> We note a connection between our setting and the problem of hypothesis selection (i.e., distribution learning with respect to the total variation metric), which may offer some useful intuition. Consider the special case of our model with a single input: here, each hypothesis corresponds to a subset of labels, and the learning problem specializes exactly to a hypothesis selection problem where both the target distribution and all distributions in the hypothesis class are uniform over subsets. Determining the optimal multiplicative factor such hypothesis selection problems was posed as an open question in the book by Devroye and Lugosi [1], and was only recently fully resolved [2]. This connection suggests that identifying the optimal factor in our setting may similarly be challenging, yet potentially approachable by leveraging techniques from the study of density estimation and hypothesis selection. We believe this is an interesting direction for future research.
>
> [1] Devroye and Lugosi, “Combinatorial Methods in Density Estimation,” Springer, 2001.
>
> [2] Bousquet, Kane, and Moran, “The Optimal Approximation Factor in Density Estimation,” COLT 2019.

---

> > ### Comment · Reviewer_PCHe · 2025-08-05
> >
> > Thanks for providing the potential solution for solving my question. I agree this direction has many potential follow-up works. I keep my score and tend to accept this paper.

---

### Decision · Program_Chairs · 2025-09-17

**Decision:**

Accept (poster)

**Comment:**

This paper makes two key contributions. First, it provides a corrective result by showing that ERM provably fails under one-sided feedback, ruling out an entire line of approaches that would otherwise mislead future research. Second, it introduces a constructive Pareto-loss perspective, giving optimal sample complexity bounds, and thereby guiding how future algorithms should be designed to balance precision and recall in this setting.

Across three positive reviews (scores 5, 4, and 4), there is clear consensus on the importance of these insights for shaping future work. Reviewers recognized the significance of establishing ERM’s failure and of formalizing Pareto trade-offs, and all agreed that the theoretical results are rigorous and impactful. The concerns raised by these reviewers focused on clarity, related work, or the absence of experiments, but none questioned the validity of the results, and all maintained positive scores.

The remaining reviewer gave a score of 2, concerning on the lack of experiments and practical algorithms as the main reason for rejection after rebuttal. While they recognized the theoretical contributions, they did not consider them sufficient in isolation. This reflects a mismatch of expectations: the paper is explicitly a theoretical contribution, and NeurIPS has long valued fundamental theoretical insights even without empirical validation. No reviewer identified flaws in the theory itself after rebuttal. Given the solid positive consensus from three reviewers and the potential lasting impact of the insights provided, I support acceptance despite the borderline mean score.